# Silicone v1.0.0: an open-source Python package for inferring missing emissions data for climate change research

Robin D. Lamboll[1], Zebedee R. J. Nicholls[2,3], Jarmo S. Kikstra[4], Malte Meinshausen[2,3], Joeri Rogelj[1]

[1]Grantham Institute for Climate Change and the Environment, Imperial College London, UK
[2] Australian-German Climate and Energy College, The University of Melbourne, Parkville, Victoria, Australia
[3] School of Earth Sciences, The University of Melbourne, Parkville, Victoria, Australia
[4] International Institute for Applied Systems Analysis, Laxenburg, Austria

*Correspondence to*: Robin Lamboll (r.lamboll@imperial.ac.uk)

**Abstract.** Integrated assessment models (IAMs) project future anthropogenic emissions which can be used as input for
climate models. However, the full list of climate-relevant emissions is lengthy and most IAMs do not model all of them. Here we present Silicone, an open-source Python package which infers anthropogenic emissions of unmodelled species based on other reported emissions projections. For example, it can infer nitrous oxide emissions in one scenario based on carbon dioxide emissions from that scenario plus the relationship between nitrous oxide and carbon dioxide emissions found in other scenarios. Infilling broadens the range of IAMs available for exploring projections of future climate change, hence
Silicone forms part of the open-source pipeline for assessments of the climate implications of IAM scenarios, led by the Integrated Assessment Modelling Consortium (IAMC). This paper presents a variety of infilling options and outlines their suitability for different cases are discussed. We recommend certain infilling techniques as the good defaults, but emphasise that considering the specifics of the model being infilled will produce better results. We demonstrate the package's utility with three examples: infilling all required gases for a pathway with data for only one emission species, splitting up a Kyoto
emissions total into separate gases and complementing a set of idealised emissions curves to provide a complete, consistent emissions portfolio. The code and notebooks explaining details of the package and how to use it are available on GitHub (https://github.com/GranthamImperial/silicone). The repository with this paper's examples and uses of the code to complement existing research is available at https://github.com/GranthamImperial/silicone_examples.

## 1. Introduction

### 1.1. General context and problem setting

Integrated assessment models (IAMs) are scientific modelling tools that integrate knowledge from different academic disciplines with the aim to explore and inform policy decisions (Clarke et al., 2014; Rogelj et al., 2018a; Weyant, 2017). They are widely used in climate change research to combine insights from energy, economy, agricultural, and natural sciences, with the aim of creating scenarios that explore how societal decisions can affect projected greenhouse gases and other emissions, as well as their related climate outcomes (Clarke et al., 2014; Huppmann et al., 2018; Riahi et al., 2017; Rogelj et al., 2018b).

However, IAMs do not always exhaustively represent all possible processes or sources of climate-relevant emissions. Thus, many IAM scenarios lack projections for some climate forcers, be it specific greenhouse gas emissions or aerosol precursors. A complete set of these climate forcers is required to accurately estimate the overall climatic effects of a given scenario (Meinshausen et al., 2011; Smith et al., 2018), as a large number of supposedly minor emissions may collectively exert a significant radiative forcing (Meinshausen et al., 2017; O'Neill et al., 2016).

Scenarios that only report a limited set of greenhouse gases or climate forcers thus must be complemented by estimated evolutions of missing emissions derived without further economic analysis. We term this estimation 'infilling'. If no infilling is attempted, the unevaluated emissions would effectively be considered zero, which would clearly create systematic biases and potential artefacts in the projected temperatures. Depending on the radiative forcing of the species in question, this bias may be positive or negative, so infilling with zeros would not necessarily be a conservative choice. Most earlier studies overcame this problem in one of two ways: with expert-based ad-hoc decisions on how to adequately fill-in missing species (Schaeffer et al., 2015); or by assuming that a pathway will occur at the same quantile for each set of emissions in a particular year, although the quantile can vary over time (Gütschow et al., 2018; Meinshausen et al., 2006; Nabel et al., 2011). However, the former clearly does not scale easily to larger databases (because making ad-hoc decisions for a thousand scenarios requires a significant time input), and the latter approach, termed the "Equal quantile walk" (EQW), ignores trade-offs and specific relationships between emission species resulting from how competing technologies, behaviours and industrial practices result in different emissions. A few alternative approaches have been used recently: for instance, using the pathway with the smallest mean-squared distance over all time was used in (Robiou du Pont and Meinshausen, 2018). This works well for large databases containing similar paths, but is less reliable for smaller databases or for paths with an unusual behaviour over time. A more sophisticated "Generalized Quantile Walk" technique can capture the effect of trade-offs and was recently introduced in section 3.8.1 in (Teske et al., 2019), involving quantile regression between a lead variable (fossil $CO_2$ emissions) and other gases for every individual year. Unfortunately, the implementation there did

not consistently guarantee that higher quantiles resulted in higher emissions, and has not been followed up with any peer-reviewed work that does so. A tool for infilling was provided with (Rogelj et al., 2014) using a cubic spline between specific points in a small database, however this type of infiller behaves chaotically when applied to large databases incorporating many different models. It was also coded in Excel, limiting the ease of open-source development.

Here we present a new toolbox of methods to address these recurring infilling challenges in the climatic assessment of socioeconomic emissions scenarios. The toolbox introduces new approaches as well as building on and combining previous approaches. The codebase is a significant improvement compared to existing options in terms of flexibility, applicability, reproducibility and versatility.

## 1.2. The aim of Silicone

Silicone is a Python package designed to enable users to expand scenario projections of a limited set of climate forcers to a broader set required for a sensible climate assessment. In essence, its methods are grounded in a comparison of the co-evolution of anthropogenic emissions in scenarios that are readily available in the literature (Huppmann et al., 2018; Riahi et al., 2017; Rogelj et al., 2018b). Silicone aims to provide IAM teams that do not represent all individual climate forcers with robust methods to complement their model output and facilitate a climatic assessment of their work. Furthermore, Silicone also aims to provide geoscience researchers with a tool to easily develop stylized, yet internally consistent future emission pathways of the most important climate forcers. It can also estimate or calculate missing emissions from particular sectors. Notebooks describing how to use these tools are available on the accompanying GitHub repository (Lamboll et al., 2020b) and the formal documentation is available at (Lamboll et al., 2020a). Additional examples of using Silicone for the specific situations outlined below are included in a separate GitHub repository (Lamboll, 2020). The package is open-source and intended to allow groups to write their own infilling methods if desired. Users and collaborators are encouraged to add any such developments to the codebase via GitHub.

Silicone is compatible with a suite of Python tools that make up the IAM climate assessment pipeline developed under the umbrella of the Integrated Assessment Modelling Consortium (IAMC). The compatibility with these tools allows us to load, manipulate and save files using a common file format. The pipeline is based around the pyam package (Gidden and Huppmann, 2019), specifically its IamDataFrame class, which Silicone makes extensive use of. Pyam dataframes easily convert from and to widely-used pandas dataframes, which pyam and Silicone also use internally (McKinney, 2011). The pipeline also includes tools to harmonise (i.e., correct projection made in the past to match now-known emissions) (called aneris, (Gidden et al., 2018a)) before infilling and to pass the complete projections to climate simulators. The estimation of climatic impact is performed by OpenSCM, (Nicholls et al., 2020), which is compatible with the data structure of the pipeline. This pipeline is being developed in support of the IAM community and the IAM scenario assessment for the forthcoming Sixth Assessment Report of the Intergovernmental Panel on Climate Change (IPCC AR6) in particular.

This paper is structured as follows: the *Methods* section presents an overview of the different infiller methods, then goes through the infiller techniques in precise and mathematical detail. In *Results*, we present our analysis of emissions projections in the SR1.5 database. This includes correlation statistics of the database, and how well Silicone reproduces one entry in the database given the other entries. We use this to draw conclusions on the implications for using Silicone on unknown data. In *Use Cases*, we present three examples of using Silicone for infilling a pathway with limited information, splitting up an aggregate basket of emissions and infilling stylised emissions trajectories. We end with a summary of our paper.

## 2. Methods

Silicone takes a database that contains data for at least two emissions species (this database is referred to as the 'infiller' database) and derives a relationship between these timeseries. It then applies that relationship to a second database (the 'target' database), which does not have any data for one of the emissions species in the infiller database. For example, based on an infiller database of $CO_2$ and $N_2O$ emissions, Silicone could then derive $N_2O$ emissions compatible with the $CO_2$ emissions in a less complete target database. In all cases, the infillers will perform best if the target data comes from a scenario that is socioeconomically similar to scenarios found in the infiller database. The performance of most crunchers can be improved by filtering out scenarios that are known to assume radically different characteristics like population number before infilling, provided that comparable emissions statistics can be found in the remaining database.

Silicone offers a range of tools that apply methods for doing this infilling which are appropriate in different circumstances, depending on the amount of complete data and how much we know about the narrative behind our emissions. These tools are referred to as 'crunchers'. Each of these crunchers takes a 'lead variable', found in both the infiller and target databases, and uses it to infer the value of a 'follower variable', found only in the infiller database (hence missing in the target database). There are also several tools for easily infilling multiple variables, called 'multiple infillers'. These may have multiple follower or lead variables.

### 2.1. Methods overview

### 2.1.1. Cruncher guide

There are currently seven types of cruncher. These are outlined in Table 1 below. A flow chart to guide the choice is shown in Figure 1. There is also a series of notebooks with examples of how to use them all in the main GitHub repository (Lamboll et al., 2020b).

### 2.1.2. Multiple Infiller and Aggregation Tools Guide

Multiple infillers are for cases where there are relationships between multiple lead or follower values that need to be considered at the same time. They allow less tailored approaches to infilling but can ensure that the infilling is faster or more consistent than infilling each of the variables separately. These are outlined in Table 2.

### 2.2. Mathematical detail

Notebooks presenting benefits and risks of each cruncher type can be found in the Silicone GitHub (Lamboll et al., 2020b) and may be useful to have as examples when analysing the work below, as well as demonstrating how to use them.

There are two main classes of infillers: those based on the ratios between two emission pathways and those based on the absolute emission values in the infiller database. If the results are to be harmonised, then harmonising both the infiller and target data before infilling is required for improved consistency (otherwise infilling depends on outdated data). Absolute value infilling techniques preserve harmonisation, however ratio-based approaches do not necessarily, and may need harmonisation again afterwards.

The ratio-based approaches are better for cases where the lead values to be infilled are outside the range in the infiller database and we expect the emissions to scale with each other. For instance, if we are infilling one incomplete combustion product based on another, or splitting up aggregated emissions into their components. However, care needs to be taken when infilling emissions that are non-negative using a lead value that may be of any sign, for example $CO_2$ emissions. In that case, the ratio method might produce values for the target emissions that are unsupported by any available evidence. Singular behaviour may also be encountered when the lead data is close to zero in the infiller database. The different crunchers present different ways to estimate the ratio to use.

The absolute value-based techniques infill with values derived from the absolute data found in the infiller database, or linear combinations of them. This means that they will always return values within the range spanned by the infiller database. This is most appropriate for processes where we have a greater number of cases, preferably with both larger and smaller lead emissions in the infiller database or where we expect the follower emissions to be strongly bounded rather than increasing in line with other variables. They may be considered more stable and more conservative. The quantile rolling windows (QRW) cruncher can be used in either ratio or absolute (non-ratio) mode, the absolute mode being the default.

As one final detail, we discuss the data model which is assumed by Silicone. Silicone is built around the pyam package (Gidden and Huppmann, 2019). As a result, it assumes that all input data is provided in a particular structure. The structure includes the model which created the timeseries, the scenario with which the timeseries is associated (e.g. a high BECS 1.5 degree scenario), the region the emissions occurs in and the unit of the data (full details available at https://pyam-

). Accordingly, Silicone is able to work on specific subsets of models (e.g. only the MESSAGE model) or subsets of scenarios (e.g. all SSP1-like scenarios). We therefore follow the pyam convention and refer

to a "model/scenario combination" to mean a single projected world, that in some contexts might be called a "scenario".

Pyam dataframes assign values to variables as a function of different models, scenarios, regions and times. All methods work on databases with only a single region at a time, although the region can be different between the infiller and target databases.

### 2.2.1. Ratio infilling methods

These methods all firstly estimate the ratio of the lead variable to the follower variable. In all cases, we first determine the ratios, written as $R(t)$ at time $t$. Once these have been calculated, the follower value in the target database, $E_f(t)$, is valued as

$$E_f(t) = R(t)E_l(t), \tag{1}$$

where $E_l(t)$ is the lead value in the target database.

**Constant ratio and latest time ratio crunchers**

'Constant ratio' and 'latest time ratio' methods both use the same ratio for all infill times, $R(t) = R$. With the 'constant ratio' method, the ratio must be given as an input parameter. The 'latest time ratio' method uses the ratio between the mean follower data in the infiller database (we denote this database with lower-case, $e_f$) and the value of the lead variable in the target data ($E_l$), both values evaluated at the latest time for which there exists follower data in the infiller database, $t_{last}$. The

mean is taken over all infiller data at that time. This is designed for the case where we have estimates only up until some time, after which it stops – for instance, if we have no projections for some new HFC emissions, but have historic measurements for recent years. This gives us the equation

$$R = \frac{\langle e_f(t_{last}) \rangle}{E_l(t_{last})}, \tag{2}$$

where the angular brackets mean taking the (algebraic) mean with equal weighting for all estimates (typically historical

estimates) at that time, and with a lower case, $e_f(t)$ represents the follower values in the database at time $t$. This ensures that at $t_{last}$, all infilled data will fulfil

$$E_f(t_{last}) = R * E_l(t_{last}) = \langle e_f(t_{last}) \rangle. \tag{3}$$

Time-dependent ratio cruncher

The 'time-dependent ratio' is appropriate for when there is some data in the infiller database for all times, and allows the

ratio to vary with time. The ratio used is

$$R(t) = \frac{\langle e_f(t) \rangle}{\langle e_l(t) \rangle}. \tag{4}$$

Optionally, the averaging can be taken only over model/scenario cases where the sign of the lead variable is the same in both the infiller and target case – this will guarantee that the infilled value takes the same sign as that of follower values in the database. It will produce an error if there is no data with the required sign. This cruncher has a useful conservativity property (with or without the sign restriction): if in every scenario averaged over, the emissions of several substances sum to another substance, e.g. if $e_1 = e_2 + e_3$, then $\langle e_1 \rangle = \langle e_2 \rangle + \langle e_3 \rangle$. It then follows that

$$1 = \frac{\langle e_2 \rangle}{\langle e_1 \rangle} + \frac{\langle e_3 \rangle}{\langle e_1 \rangle}, \tag{5}$$

the right-hand side of which we can identify as the two $R(t)$ values of using formula (3) twice for different followers. This means when the aggregate is the lead and the components are followers, the sum of the two ratios is one, so we can use this infiller to break an aggregate value into components and know that the total is conserved. This relationship generalises to any number of components, still holds when emissions can be negative, and is irrespective of whether the averaging includes all values or only those where the lead has a particular sign.

This cruncher is the foundation for the 'decompose collection with time-dependent ratio' multiple infiller. This relies on all scenarios having values for all of these variables, so misses out cases which do not have one of the constituents or only reports at some of the required times, unless the override option "only_consistent_cases" is set to False. It always constructs a new, consistent version of the aggregate variable in case different modellers used different conversion factors in the infiller database.

**Quantile rolling windows cruncher**

The 'quantile rolling window' method may be applied in ratio mode, in which case we calculate $R(t)$ by first calculating the ratio for each scenario,

$$r_s(t) = \frac{e_f(t)}{e_l(t)}, \tag{6}$$

then following the calculation in the absolute value section, using this instead of $e_l$. This method finds quantiles of the ratio in the infiller database at set points along the range of lead values in the infiller database.

**2.2.2. Absolute value infilling methods**

**RMS closest cruncher**

The 'RMS closest' cruncher filters the infiller database for models with data at all the times found in the target data. It then ranks models and scenarios by the root mean squared (RMS) difference between the lead data in the infiller and target database, with the average being taken over all timeslices. It returns the follower data from the scenario/model combination

with the smallest RMS difference: the formula is $E_f(t) = e_{f,i}(t)$, where the subscript $i$ refers to the model/scenario case that minimises

$$\sum_t \left( E_l(t) - e_{f,i}(t) \right)^2. \tag{7}$$

In the case of a draw, the value that occurs earlier in the infiller database will be used. This is the only infiller that is not time-independent, i.e. changing the value of the lead at one time may result in different outputs at other times.

**Linear interpolation**

The 'linear interpolation' method constructs an (unsmoothed) linear interpolator function between all lead and follower points in the infiller database at a given point in time. It is similar in concept to the cubic spline interpolator used in (Rogelj et al., 2014). The equation for our case is

$$E_f(t) = +, \tag{8}$$

where subscript $<$ or $>$ signs indicate the model/scenario combination with lead values immediately below or above the target lead value at that time. If multiple points have exactly the same lead value, the average follow value is used. The follower value returned is then the interpolated value for the target lead. The 'Interpolate specified scenarios and models' cruncher filters for scenarios and models that match a given text string before performing the same action of the linear interpolation cruncher.

**Quantile rolling windows cruncher**

The 'quantile rolling windows' cruncher, applied with the default option 'use_ratio=False', infills the values based on interpolating between the required quantile of the follower variable. This is calculated at fixed points across the range of lead values in the infiller database for each time. The process is identical to the above discussion where 'use_ratio' is True, except using the actual follower values instead of the ratios between lead and follow. It is inspired by the Generalized Quantile Walk approach in section 3.8.1 of (Meinshausen and Dooley, 2019). An illustration of the idea behind this cruncher is shown in Figure 2. For each time in the infiller database, it splits the range of lead values into $n_{windows}$ points (defaults to 10) with values $e_p$, including the highest and lowest values. For each window, the weightings of each point are given as

$$w_p(e_l(i)) = 1/(1 + \left( (e_p - e_l(i)) \big/ d_l \right)^2), \tag{9}$$

where $d_l$ is the decay length, which defaults to half the separation between $e_p$, and $i$ the label for which model/scenario we are investigating. Increasing the decay length will reduce the weight difference between points, so the rolling window becomes wider and more even, with the limit case of calculating quantile $q$ of all data for large $d_l$. Amongst other things, this is a clear improvement over the Generalized Quantile Walk approach, as the latter uses equal weights within a fixed window

of a certain fraction of the infiller database's lead values in a certain year. These values are then normalised so that $\sum w_p = 1$ and sorted into ascending order by $e_f$. The follow value at quantile $q$, evaluated at lead point $e_l(j)$, is where the quantile equals the sum of weights of all smaller $e_f$ plus half the weight of $e_f(j)$ itself. Note that we sum over smaller *follower* values, but the weighting is determined by the *lead* values:

$$q\big(e_l(j)\big) = \sum_{e_f(i)<e_f(j)} w_p\big(e_l(i)\big) + \frac{w_p(e_l(j))}{2}. \tag{10}$$

Quantiles between these are evaluated by linearly interpolating this relationship. We are usually interested in the case where $q = 0.5$. To infill a point at $E_l$, we interpolate between the known points $e_p$. Quantile crossing is not possible in this framework because at any given evaluation point higher quantiles cannot have lower values, and only linear fits between these points are used.

**Equal quantile walk**

The equal quantile walk calculates the quantile of the lead value at each time (Meinshausen et al., 2006). This is zero for values below the database minimum, one for those above the database maximum and the fraction of infiller data smaller or equal to this value otherwise. We interpolate between neighbouring values in the infiller data to find the fraction that would match the target value exactly. We then apply the same logic to calculate the appropriate value for the derived quantile of the follower data.

### 2.3. General limitations

Note that all of the methods listed above are purely statistical in nature: if the scenarios in the infiller database are fundamentally different from those in the target database, different relationships are likely and the validity of the results is poor. The adequate use of Silicone requires users to select an infiller database most appropriate for each respective application. Using Silicone with an infiller database that has itself been infilled may distort the model democracy of the results. Note also in version 1.0.0 of Silicone, all methods take only a single lead value, although forthcoming work will add the capacity to use multiple lead values to some crunchers. This will improve the ability to resolve more complex relationships, since it is possible for very different worlds to have similar emission trends in one emission without being similar in other emissions.

# 3. Results

## 3.1. Rank Correlations

The infilling method is important. However, equally important is the choice of lead variable. The best choice is where there is a causal link between the lead and follower variable, particularly if there is a clear understanding of the implications of this link for the relative behaviour of the two variables, for instance black carbon and carbon monoxide are both produced by incomplete combustion. In most cases, there is no such certainty, and the best choice is then to find the lead variable with the best predictive power. We estimate this by the Spearman's rank correlation coefficient, a measurement of the monotonicity of the relationship between the two variables. In cases where this value is low, we anticipate the need for higher effort to select relevant cases from the infilling database. We use the data from the IPCC Special Report on Global Warming of 1.5 ℃ (Huppmann et al., 2018) as our database of scenarios and compare the correlations between the different variables. The Silicone package has a function in the statistics section called 'calc_all_emissions_correlations', which will produce tables of both the Spearman (rank) and Pearson correlation coefficients, calculated separately for each year requested and also the time-averaged magnitude of the correlations. Since there is no reason to expect the relationships between variables to be linear, we will focus on the rank correlation in this analysis. We also plotted the relationships between $CO_2$ and all other variables (using the plotting function in the Silicone examples github) to check that there were no obvious cases of a non-monotonic relationship. All the crunchers work just as well with negative trends as with positive, so the sign of the correlations is not relevant for considering goodness of fit. Using this tool, we can calculate the decadal-averaged magnitude of the rank correlation coefficient, found in Table 3. We also calculate the variation of this value with time, and in cases where this exceeds 0.03 (chosen to highlight only extreme cases), colour the cells blue. This is to indicate cases where more care needs to be taken to ensure that values are representative for the times of interest.

The immediate observation from the study of absolute rank correlations is that there is no clear, overall best infiller gas. $CH_4$ has a slightly higher average than other emissions and is reported by most models. $CO_2$ is reported by all models and has the second highest correlation, however this is somewhat inflated by having two of its constituents listed separately (Agriculture, Forestry and Other Land Use (AFOLU) and Energy and Industrial Processes, a similar concern can be raised about F-gases). Generally, $CO_2$ and $CH_4$ are therefore the best choices for a 'default lead variable'. However, there are some specific cases where the correlations are low, and much better choices could be made.

There is a cluster of emissions species, specifically black carbon, organic carbon and carbon monoxide, that correlate well with each other but less well with other emission pathways. Physically, these relate to incomplete burning, and are best infilled using each other. The F-gases, $SF_6$, hydrofluorocarbons (HFCs), and perfluorinated compounds (PFCs) also

primarily relate to each other. Many models report F-gas emissions as a basket. Infilling these should best be done by splitting the F-gas basket into its constituents. Otherwise the default infillers, $CO_2$ and $CH_4$, should do reasonably.

## 3.2. Reconstructing data

The choice of cruncher to use in different situations will depend on the expectations about the specific emissions in question. However, in cases where there are no clear expectations, it is good to have a default. In this section we assess to which degree the cruncher reproduces the follower data from one model and scenario given the lead data from that case and all data from all the other model/scenario combinations in the SR1.5 database. We try this with both $CH_4$ and $CO_2$ as our lead variables. We use the crunchers that are designed for use on complete datasets with only default settings: QRW (default settings mean in absolute mode and for the 0.5 quantile), RMS closest, EQW, time-dependent ratio and linear interpolation. Interpolate selected model behaves identically to linear interpolation with default settings and is not treated separately here. We perform the infilling for each model/scenario combination, for each decade from 2020 to 2100, and report the root mean squared difference between the original value and the infilled value, normalised by the standard deviation in the follower value in the database at that time ($\sigma$), i.e. $\langle \sqrt{\langle \left( \frac{E_{f,inf} - E_{f,act}}{\sigma} \right)^2 \rangle_i} \rangle_{decade}$, with the subscript text $inf$ indicating that the value is infilled, $act$ indicating actual and $i/decade$ indicating averaging over model/scenario cases or decades. These results are found in tables 4 and 5. Given the definition of standard deviations, values larger than one would indicate that this infiller is worse than simply using the mean value in the database.

We see with this fairly large infiller database that for both $CO_2$ and $CH_4$ the approach that generates follower pathways most similar to those removed from the initial scenarios (i.e. the smallest errors) is the RMS technique, with the QRW technique being the next smallest. Linear interpolation without smoothing is expected to produce a noisy fit when given a large infiller dataset, so its performance is unsurprisingly worse. The Equal Quantile Walk (EQW) performs similarly poorly, due to effectively ignoring the relationship between the lead and follower data. The time-dependent ratio method is worst of all – its errors are potentially unbounded and for $CO_2$ the average error far exceeds one. To determine the appropriate statistics to apply on the errors, we first perform a Shapiro-Wilk test to detect any non-Gaussian aspect for the error distribution (details can be found in the "statistics_for_paper" notebook of the examples github repository). This indicated that the distributions are statistically significantly non-Gaussian, for several crunchers when analysed separately and most clearly as an aggregate. We will therefore use non-parametric tests where possible. The small differences in rank between $CH_4$ and $CO_2$ manifest in slightly lower values for $CH_4$. Performing a Wilcoxon's t-test on the results indicates that this result is statistically significant for the data as a whole (relative t-test t-statistic 376, p = 0.00007), although when considering each of the crunchers individually, only the RMS closest and time-dependent ratio crunchers are significantly better with $CH_4$ than $CO_2$ (p-values for time dependent ratio = 0.012, QRW = 0.48, RMS Closest = 0.041, linear interpolation = 0.060, EQW = 0.39). We

therefore conclude that using either $CO_2$ or $CH_4$ as the default will produce the most reasonable results when using one infiller species, with $CH_4$ performing slightly better, while also generally having a slightly lower availability of data.

We perform similar pairwise Wilcoxon t-tests on the results of different crunchers, and find that the ordering of mean errors, (RMS closest < QRW < Linear interpolation ≈ EQW < Time dependent ratio) are all statistically robust. The p-values are < 0.01 for almost all pairs except linear interpolation and EQW, which are much greater than 0.1, whether the comparison uses $CO_2$ or $CH_4$ lead data, or all data combined. The one pairwise exception to this is time-dependent ratio and EQW for $CH_4$, which has only $p=0.028$, though the values for other combinations still have $p < 0.01$.

We stress that this does not always mean that the RMS closest technique is the best default, as it makes the assumption that the pathway being infilled is similar to a whole pathway found in the database. The advantage of the quantile rolling windows technique is its choice of conservativity – for example that it tends to produce values more towards the median value if the default 0.5 quantile is used – and time-independence, whereas RMS closest is better at reconstructing the data and has better consistency over time. Linear interpolation, EQW and time-dependent ratio are best used in cases where there is a large degree of knowledge about the expected relationship between variables.

## 4. Use cases

Data in the Silicone examples package relies on the IAMC 'pyam' open-source software data structure (Gidden and Huppmann, 2019) and fits into the IAMC scenario assessment pipeline prepared in support of the IPCC AR6 literature assessment.

As part of the pipeline, emissions projections are also harmonised, i.e. modified to be consistent with known historical emissions in a smooth way (Gidden et al., 2018a). The Silicone process is assumed to be part of the IAMC pipeline after harmonisation, as the harmonisation process will potentially differently affect the target and infiller data, resulting in inconsistencies. All infiller options except latest time ratio are designed such that if both the data being infilled (the 'target data') and the data drawn on for infilling ('infiller data') are harmonised, the result must also be harmonised, so there is no need for harmonisation again after infilling. (Latest time ratio only preserves the harmonisation of the last timepoint in the infiller database.) The infilled results can then be run via climate models, most easily via the OpenSCM package (Nicholls et al., 2020).

We now demonstrate several uses of the package for specific purposes. The notebooks demonstrating the steps for these calculations can be found in the *Silicone_examples* GitHub repository (Lamboll, 2020), along with several other use-cases.

### 4.1.1. Infilling the IMAGE model POEM scenario B

To demonstrate the uses of this package alone, we will apply the methods directly using unharmonised data in the SR1.5 repository (Huppmann et al., 2019) to infill the emission pathways of the POEM scenario B from the AR5 database (Clarke

et al., 2014). The POEM scenarios only report $CO_2$ from certain sources and are thus an excellent use case. The crunchers are all used via the multiple infiller, "infill_all_required_emissions_for_openscm". No active decisions are taken except to use the SSP2 scenarios from the MESSAGE model for the specified model interpolation. The choice of SSP2 in this case is ultimately arbitrary but supported by POEM scenario B being fairly middle-of-the-road and usually fitting in the SSP2 range.

The choice of MESSAGE model is because this is the marker model for SSP2 (Riahi et al., 2017). Other POEM scenarios would need different ranges of scenarios for infilling.

We see from Figure 3 that the linear interpolation model (without filtering the database) provides a chaotic pathway, due to its value being determined only by the two points either side of it in the database, which changes semi-randomly with time and should not be used here. Although the interpolate specified model approach is also determined by only a few

model/scenario pairs because there is only data from a small number of related scenarios, the pathway is smoother and more consistent. The EQW pathway assumes a strong, direct relationship between $CO_2$ and $CH_4$ emissions which the other crunchers do not uphold at early times, although this would disappear if the data were harmonised. The other cruncher results are all fairly similar and look consistent. The RMS closest pathway is consistent by construction (and precisely overlines a point in the original database). The quantile rolling windows result also looks consistent and tends to move closer

to dense clouds of values in the infiller database. In deciding which is the best infiller to use, the RMS closest result is more consistent over time but more arbitrary in its selection of the pathway, while quantile rolling windows is more conservative in the sense of giving results closer to the median behaviour of the whole data set.

### 4.1.2. Splitting up a Kyoto Greenhouse Gases path

The Silicone package has features that can split a basket of gases into its constituents. In this example we take data from the

360 Climate Action Tracker (CAT) website (https://climateactiontracker.org/) (Climate Action Tracker, n.d.), which reports projected global emissions in terms of Kyoto gas totals. While it is possible to use this to infill all other values directly as above, the subcategories of Kyoto gas will not necessarily add up to the Kyoto gas total. Therefore, one of the multiple infillers designed for this use is preferable. The symmetric way to divide the basket into its constituent parts ($CO_2$, $CH_4$, $N_2O$ and F-gases), is using the 'decompose collection with time-dependent ratio' multiple infiller, which uses a ratio-based

technique to ensure conservation of the total amounts. Alternatively, the 'split collection with remainder' multiple infiller can estimate the fractions of $CH_4$, $N_2O$ and F-gases, then assign the remainder to $CO_2$. F-gases could be further subdivided using similar methods.

As can be seen in Figure 5, the curves that result from decompose collection are generally smooth, in spite of being separately calculated at each timepoint. It is important to ensure that the number of scenarios reported at each time are

370 consistent. In the SR1.5 database, some scenarios only report values at decadal intervals, whereas others use five-year intervals. We interpolated all models to five-year intervals to give consistent representation. In the $CH_4$ and F-gases, the

lowest orange line is clearly seen to rise discontinuously after 2060. This is the last point before the Kyoto total goes negative. To ensure that the sign of the constituents is correct, the formula only considers data from SR1.5 paths where the Kyoto total has the same sign as in the data being infilled. In this way, emissions that are unlikely to go negative like $CH_4$ are ensured positive, however their magnitude increases the more negative the aggregate is.

For this reason, the 'split collection with remainder' method produces more robust results with sign changes in the lead variable. This technique can use any cruncher, usually RMS closest or (probably non-ratio) quantile rolling windows to infill the positive values and then allow the value that may be negative ($CO_2$) to make up the rest. This produces the results seen in Figure 6. Here behaviour of all curves is fairly smooth, with no obvious features around zero-crossing points and no negative values except in $CO_2$, as expected.

### 4.1.3. Stylised trajectories

Another use of this software is to infill simple, stylised trajectories generated to explore a wide range of possibilities without detailed economic modelling. For example, Sanderson *et al.* (Sanderson et al., 2016) suggest simple formulae whereby one may construct emissions trajectories characterised by a few free variables – in this case, based on rates of transition between the RCP pathways and a long-term emissions value. They present general formulae for generating plausible total $CO_2$ pathways with several free variables. Silicone provides an alternative means of complementing such results – instead of specifying the functional forms of all emissions, you can have a few key emissions prescribed and infill the remainder using scenarios with similarities to the desired narrative. A notebook can be found in the Silicone examples github detailing the calculations and demonstrating this usage, titled "Infill_stylised_path.ipynb" (Lamboll, 2020), using data from (Riahi et al., 2011; van Vuuren et al., 2011). It shows that curves with different values in some of the parameters, termed $E_\infty$ and $\tau$, can be complemented using a number of techniques. Here we highlight the method of interpolating results from any of the SSP scenarios as implemented by variants of the MESSAGE model. As the different SSPs have different narratives, this allows the user to decide what narrative is relevant to the infilling, rather than adding more arbitrary values (Gidden et al., 2018b). An example of this output can be found in Figure 7.

### 5. Summary

In this paper we have outlined the features of the open-source Silicone package. This provides tools for complementing emissions pathways with other climate-relevant emissions through relationships found in the scenario literature. The package features several scripts for analysing data to establish the relationships between the variables in the complete infiller database, to establish the best variables to use when infilling. The values of the follower data are estimated using objects called crunchers. Notebooks describing the use of the crunchers are included in a GitHub repository

(https://github.com/GranthamImperial/silicone), which also contains full documentation. In addition, a flowchart for guiding the choice of cruncher for a given situation is included in the text. The results of Spearman's rank correlations and applying the crunchers to the SR1.5 database implied that the best default lead variables are $CH_4$ and $CO_2$, and that the best default cruncher is the root mean squared closest cruncher, followed by the quantile rolling windows cruncher. Both of these

crunchers perform significantly better at reconstructing known pathways compared to the commonly used equal quantile walk technique, although this and many other crunchers are included in the package for specific situations where they are more appropriate. Using several examples and use-cases of different infilling techniques, this paper has demonstrated that Silicone can easily be used to allow the involvement of a broader range of IAMs in making climate assessments.

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

**Author contributions**

JR initiated the research based on earlier work by MM. RDL led the code development and the mathematical translation of infiller methods. RDL and ZRJN wrote the code, JSK assisted reviewing it. JR and MM conceived infilling techniques and use cases. RDL wrote the manuscript, all authors gave comments and contributed to the final version.

**Code availability**

The Silicone code in this paper is available from the main github repository (Lamboll et al., 2020b). Code used to analyse the 510 output of Silicone is available in a second github repository (Lamboll 2020).

**Acknowledgements**

This project has received funding from the European Union's Horizon 2020 research and innovation programme under grant agreement No 820829 (CONSTRAIN). We thank Nicholai Meinshausen for useful statistics discussions.

**Table 1: A guide to crunchers. Names followed by asterisks use a ratio-based approach, i.e. they find a multiplicative factor and then multiply the target lead by this value. These crunchers do not preserve harmonisation. If the asterisk is in brackets, a ratio-based approach is optional. Otherwise, techniques all return linear combinations of values seen in the infiller database.**

| Name | Description | Use case | Pitfalls |
|---|---|---|---|
| Constant ratio* | Multiplies the lead variable by a constant (not fitted to any data) | Used when no information about the follower variable is available in any database. Mainly used for infilling with zeros. | Has no basis in the data – only used as a last resort in cases of complete uncertainty. |
| Latest time | Multiplies the lead variable | Used when no data is | No reason to assume that the |

| | | | |
|---|---|---|---|
| ratio* | by a constant fitted to a single (latest) timepoint in the infiller data. | available for most times, this generalises from the latest information we have, e.g. if only historic data is available. | relationship between emissions holds for all time. No restriction on signs of follower gas, so potential sign errors when the lead (but not follower) emissions may become negative. Sensitive to emissions trajectories with a high coefficient of variation. |
| Time dependent ratio* | Multiplies the lead variable by the ratio of the averages of the lead and follower data in the infiller database. (Note: this ratio is not the same as the average of the ratios and is more stable to inclusion of extreme ratios.) Optionally calculates this using only values with the same sign of lead emissions. | Used when two emissions should track each other, or one represents a portion of the other. | Allows arbitrarily high emissions. Can behave unexpectedly if emissions change sign, and an error is produced if emissions with this sign are not seen at the same time in the infiller database. |
| RMS closest | Finds the most similar pathway in the infiller database and uses those values. Most similar means smallest root mean squared difference between the lead values of infiller pathways and target pathway averaged over all times. | Used when behaviour at one time should strongly determine behaviour at another and continuity is needed between times. The only cruncher that does not treat each time separately. | A small change in the target data at a single timestep can result in large changes in output at every timestep. All the results returned are found exactly in the infiller database, so if that database is small, the same values are returned in many cases. Results more extreme than found in the infiller database all return the same value. |
| Linear interpolation / Interpolate specified scenarios and models | At each time, linearly interpolates between the follower values at the two nearest lead values, taking averages where multiple points have identical lead values. Interpolate specified scenarios and models filters the infiller database before applying the same technique. | Used for infilling where we have a small number of comparable models/scenarios. The required filtering gives control over the narrative used for infilling. | A small change in the target data can result in a large change in the output at the same timestep because pathways in the infiller database can be very different in follower variables for nearly identical values of the lead variable. For similar reasons, results can vary erratically between timesteps for large infiller datasets. Results more extreme than those found in the infiller database all return the same value. |

| Quantile rolling windows (QRW)/time-dependent quantile rolling windows (*) | At each time, applies a $1/(1+(\text{lead variable difference})^2)$ weighting to datapoints at equally spaced points across the infiller lead. Then calculates a specified quantile (usually the median) for the infiller follower value at these points. Can also be used in ratio mode, in which case the ratio between lead and follower in the infiller database is treated as above. Time-dependent QRW allows the quantile to be different at different times (but is computationally slower). | Can choose options to give more smoothing (less noise) or more localised behaviour (shows trends better) Allows the option to generate a distribution of outputs, not just a single optimum. Can add to the narrative through time-dependence. Ratio mode allows better infilling outside the range of the infiller data. | Using with any quantile larger than 0.5 will result in all emissions being higher, even if the lead and follower emissions anticorrelate. Results more extreme than found in the infiller database all return the same value, unless in ratio mode. In ratio mode, sign changes in the lead variable can result in follower emissions being assigned undesired negative values. |
| Equal quantile walk (EQW) | Calculates the quantile of the infiller database corresponding to the lead value in each individual year. Returns that quantile in that year of the follow value from the same database. | Conceptually simple, used by previous work. | Assumes all variables are monotonically increasing together. Results more extreme than those found in the infiller database all return the same value. |

**Table 2: Guide to aggregation tools and multiple infillers. Names followed by asterisks use a ratio-based approach, i.e. they find a multiplicative factor and then multiply the target lead by this value, if the asterisk is in brackets there are ratio-based.**

| Name | Description | Use case | Pitfalls |
|---|---|---|---|
| Aggregation tools | | | |
| Aggregate to composite values | Requires only the target database. Adds together known values to construct a consistent output (with optional weighting). | Infilling aggregate values (e.g. Kyoto gas totals) or finding remainders given aggregates and values for the other components. | Requires all information to be known already – no statistical inference, just adding. |
| Multiple infillers | | | |
| Decompose collection with time-dependent ratio* | Constructs a consistent version of the aggregate in the infiller database. Breaks a known quantity down into components, estimated by the time-dependent ratio method. | Breaking down aggregate values into their components, assuming all should be treated similarly. | Infiller scenarios which do not have values for all components at all times are ignored. Ignores the aggregate if the infiller database has inconsistency between that and the sum of reported components. Assumes direct proportionality between components and sum, which is problematic around sign changes. |
| Split | Breaks an aggregate | Breaking down aggregate | The remainder emission is not |

| collection with remainder emissions | emission into most of its separate components, with one emission type making up the remainder of the emissions. | values into their component when one emission type is much larger than the others, or may be either positive or negative | constrained, nor as precisely estimated as the other values. |
|---|---|---|---|
| Infill all required values (*) | Uses the same lead variable and cruncher to infill any gaps in emissions data. | For infilling scattered, minor gaps in a largely sound database. | Low confidence in the results being accurate as the method does not consider the specific characteristics of the data. |

**Table 3. Absolute values of Spearman's Rank correlation between emissions, averaged over the start of decades from 2020 to 2100. We use the following abbreviations: BC as black carbon, VOC as volatile organic compounds, AFOLU as Agriculture, Forestry and Other Land Use; and En & IP as as energy and industrial processes. "CO$_2$|" represents subtypes of CO$_2$. We also calculate the average of these rows, with or without the CO2 and subtypes. Cells are bold and yellow if the value in them is > 0.7 and are blue if the variance of the rank correlation between years exceeds 0.03. There is no overlap between these categories.**

| Variable | BC | CH$_4$ | CO | CO$_2$ | CO$_2$|AFOLU | CO$_2$|En & IP | F-Gases | HFC | N$_2$O | NH$_3$ | NOx | OC | PFC | SF$_6$ | Sulf | VOC |
|---|---|---|---|---|---|---|---|---|---|---|---|---|---|---|---|---|
| BC | | 0.47 | **0.75** | 0.46 | 0.37 | 0.42 | 0.23 | 0.10 | 0.40 | 0.40 | 0.58 | **0.73** | 0.41 | 0.20 | 0.48 | 0.45 |
| CH4 | | | 0.32 | **0.74** | 0.49 | **0.73** | 0.64 | 0.58 | **0.86** | 0.34 | 0.58 | 0.30 | 0.66 | 0.41 | 0.65 | 0.24 |
| CO | | | | 0.36 | 0.38 | 0.32 | 0.06 | 0.16 | 0.29 | 0.35 | 0.48 | **0.78** | 0.05 | 0.17 | 0.36 | 0.68 |
| CO2 | | | | | 0.54 | **0.96** | 0.60 | 0.57 | 0.54 | 0.30 | 0.61 | 0.24 | 0.35 | 0.22 | 0.69 | 0.37 |
| CO2| AFOLU | | | | | | 0.36 | 0.27 | 0.40 | 0.53 | 0.36 | 0.33 | 0.34 | 0.23 | 0.21 | 0.31 | 0.20 |
| CO2| En & IP | | | | | | | 0.58 | 0.51 | 0.50 | 0.25 | 0.61 | 0.17 | 0.32 | 0.18 | 0.69 | 0.36 |
| F-Gases | | | | | | | | **0.91** | 0.57 | 0.19 | 0.50 | 0.10 | **0.90** | **0.77** | 0.60 | 0.12 |
| HFC | | | | | | | | | 0.46 | 0.11 | 0.30 | 0.14 | **0.71** | 0.68 | 0.36 | 0.23 |
| N2O | | | | | | | | | | 0.44 | 0.46 | 0.30 | 0.65 | 0.40 | 0.49 | 0.17 |
| NH3 | | | | | | | | | | | 0.23 | 0.39 | 0.10 | 0.05 | 0.23 | 0.25 |
| NOx | | | | | | | | | | | | 0.22 | 0.53 | 0.26 | **0.76** | 0.39 |
| OC | | | | | | | | | | | | | 0.20 | 0.11 | 0.19 | 0.41 |
| PFC | | | | | | | | | | | | | | **0.77** | 0.46 | 0.16 |
| SF6 | | | | | | | | | | | | | | | 0.26 | 0.24 |
| Sulfur | | | | | | | | | | | | | | | | 0.46 |
| VOC | | | | | | | | | | | | | | | | |
| Average | 0.43 | 0.53 | 0.37 | 0.50 | 0.36 | 0.46 | 0.47 | 0.42 | 0.47 | 0.27 | 0.46 | 0.31 | 0.43 | 0.33 | 0.47 | 0.32 |
| Average, no CO$_2$ | 0.43 | 0.50 | 0.37 | 0.46 | 0.34 | 0.43 | 0.47 | 0.40 | 0.46 | 0.26 | 0.44 | 0.32 | 0.47 | 0.36 | 0.44 | 0.32 |
| # scenarios | 389 | 412 | 353 | 414 | 412 | 414 | 368 | 108 | 411 | 345 | 363 | 363 | 180 | 191 | 412 | 345 |

**Table 4: Root mean squared error in reconstructing known data using different crunchers, with $CO_2$ as the lead variable, normalised by the standard deviation at that time.**

| Species | Time dependent ratio | QRW | RMS Closest | Linear Interpolation | EQW |
|---|---|---|---|---|---|
| BC | 1.763 | 0.734 | 0.668 | 1.021 | 0.921 |
| $CH_4$ | 0.774 | 0.460 | 0.392 | 0.520 | 0.500 |
| CO | 2.236 | 0.804 | 0.764 | 1.049 | 1.006 |
| F-gases | 0.576 | 0.537 | 0.485 | 0.619 | 0.603 |
| HFC | 0.618 | 0.559 | 0.512 | 0.606 | 0.581 |
| $N_2O$ | 1.566 | 0.645 | 0.535 | 0.797 | 0.786 |
| $NH_3$ | 1.681 | 0.781 | 0.676 | 1.076 | 1.060 |
| NOx | 1.538 | 0.662 | 0.606 | 0.826 | 0.771 |
| OC | 2.062 | 0.792 | 0.706 | 1.069 | 1.112 |
| PFC | 0.649 | 0.576 | 0.441 | 0.600 | 0.764 |
| $SF_6$ | 0.754 | 0.653 | 0.499 | 0.762 | 0.809 |
| Sulfur | 0.819 | 0.570 | 0.494 | 0.658 | 0.637 |
| VOC | 2.223 | 0.812 | 0.708 | 1.056 | 1.007 |
| **Mean** | **1.328** | **0.660** | **0.576** | **0.820** | **0.812** |

**Table 5: Root mean squared error in reconstructing known data using different crunchers, with $CH_4$ as the lead variable, normalised by the standard deviation at that time.**

| Species | Time dependent ratio | QRW | RMS Closest | Linear Interpolation | EQW |
|---|---|---|---|---|---|
| BC | 1.082 | 0.729 | 0.657 | 0.971 | 0.875 |
| CO | 1.410 | 0.798 | 0.642 | 1.017 | 1.018 |
| $CO_2$ | 0.626 | 0.468 | 0.448 | 0.541 | 0.483 |
| F-gases | 0.659 | 0.565 | 0.506 | 0.657 | 0.664 |
| HFC | 0.697 | 0.593 | 0.471 | 0.669 | 0.649 |
| $N_2O$ | 0.719 | 0.457 | 0.364 | 0.497 | 0.441 |
| $NH_3$ | 1.134 | 0.756 | 0.533 | 0.958 | 1.048 |
| NOx | 0.919 | 0.680 | 0.625 | 0.823 | 0.758 |
| OC | 1.318 | 0.777 | 0.584 | 0.972 | 0.989 |
| PFC | 0.592 | 0.546 | 0.312 | 0.550 | 0.702 |
| $SF_6$ | 0.703 | 0.633 | 0.502 | 0.768 | 0.799 |
| Sulfur | 0.610 | 0.580 | 0.508 | 0.627 | 0.644 |
| VOC | 1.398 | 0.802 | 0.618 | 0.972 | 1.038 |

| Mean | 0.913 | 0.645 | 0.521 | 0.771 | 0.778 |
|---|---|---|---|---|---|

Figure 1: Flow chart suggesting how to choose the cruncher (peach oblongs) or multiple infiller (yellow oblongs) to use when infilling.

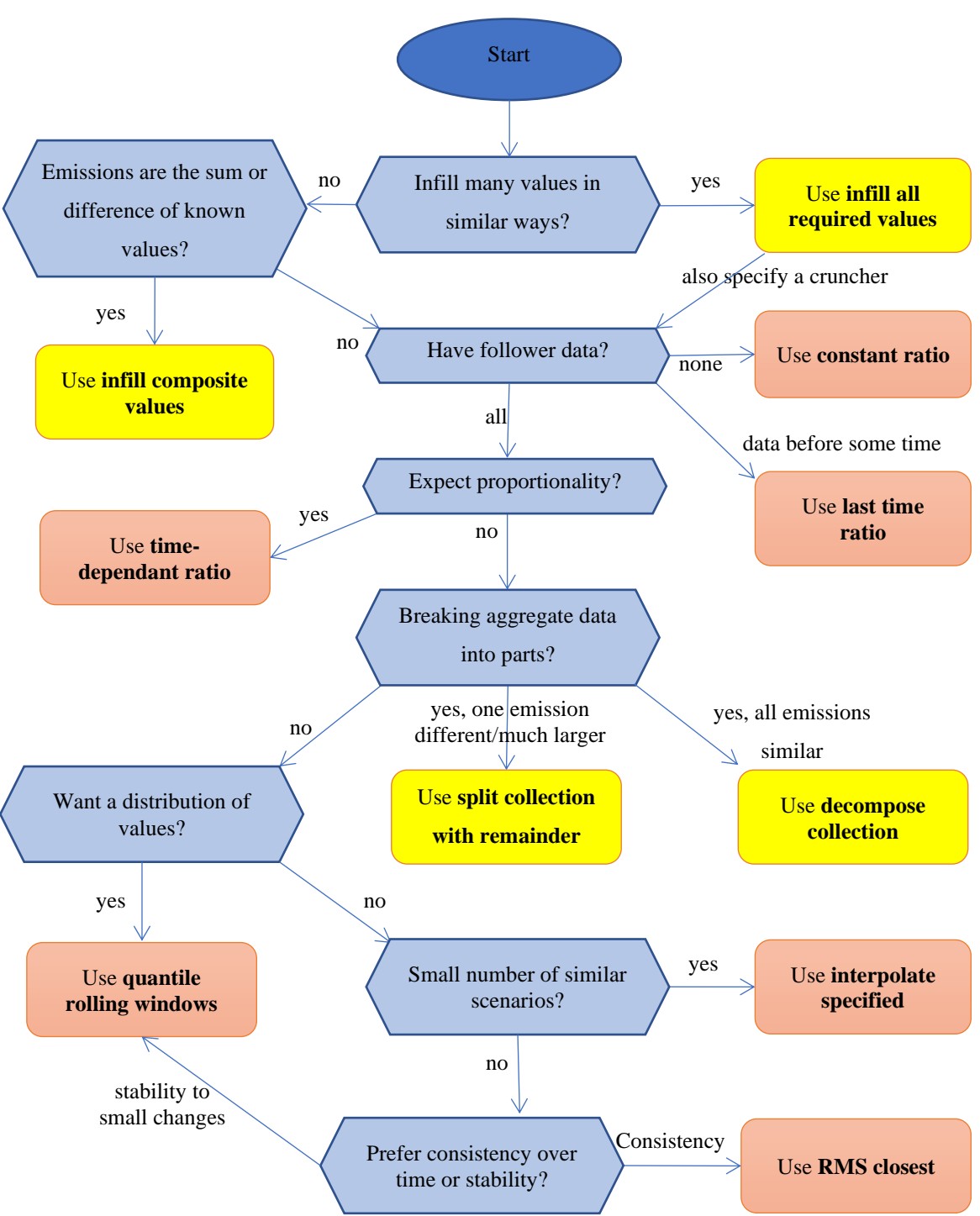

Figure 2: Schematic of how the quantile rolling windows cruncher determines the follow value to use. a) Example relationships between lead ($CO_2$) and follow ($CH_4$) variables over time. b) A number of rolling windows centers (here 5, by default 10) are drawn and a weighting function constructed for each window. It has a continuous distribution, rather than a discrete cutoff, hence the name. c) A relationship between the sum of the weights and the follow value is established and the follow value at the desired quantile is returned.

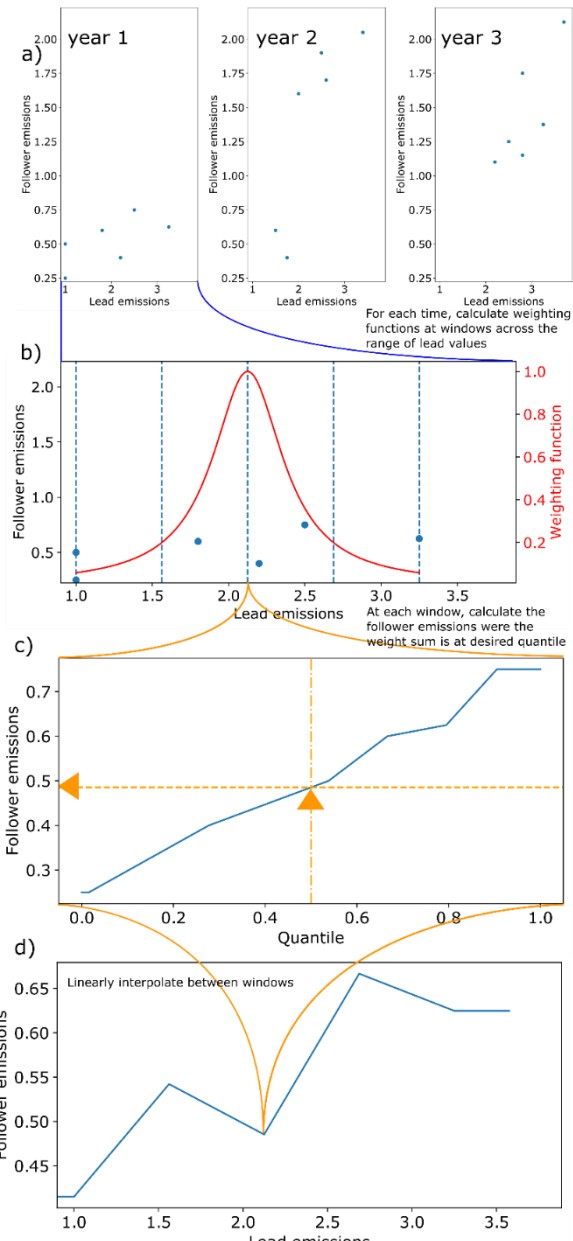

**Figure 3: Left: The POEM scenario B projection for $CO_2$ from Energy and Industrial Applications data. The fine lines represent the different timeseries in the SR1.5 database used to perform the infilling and are not included in the legend for clarity. Right: The results of interpolating this data using five different crunchers. The interpolate specified model approach used the MESSAGE**
**model and only choses scenarios based on SSP2 pathways.**

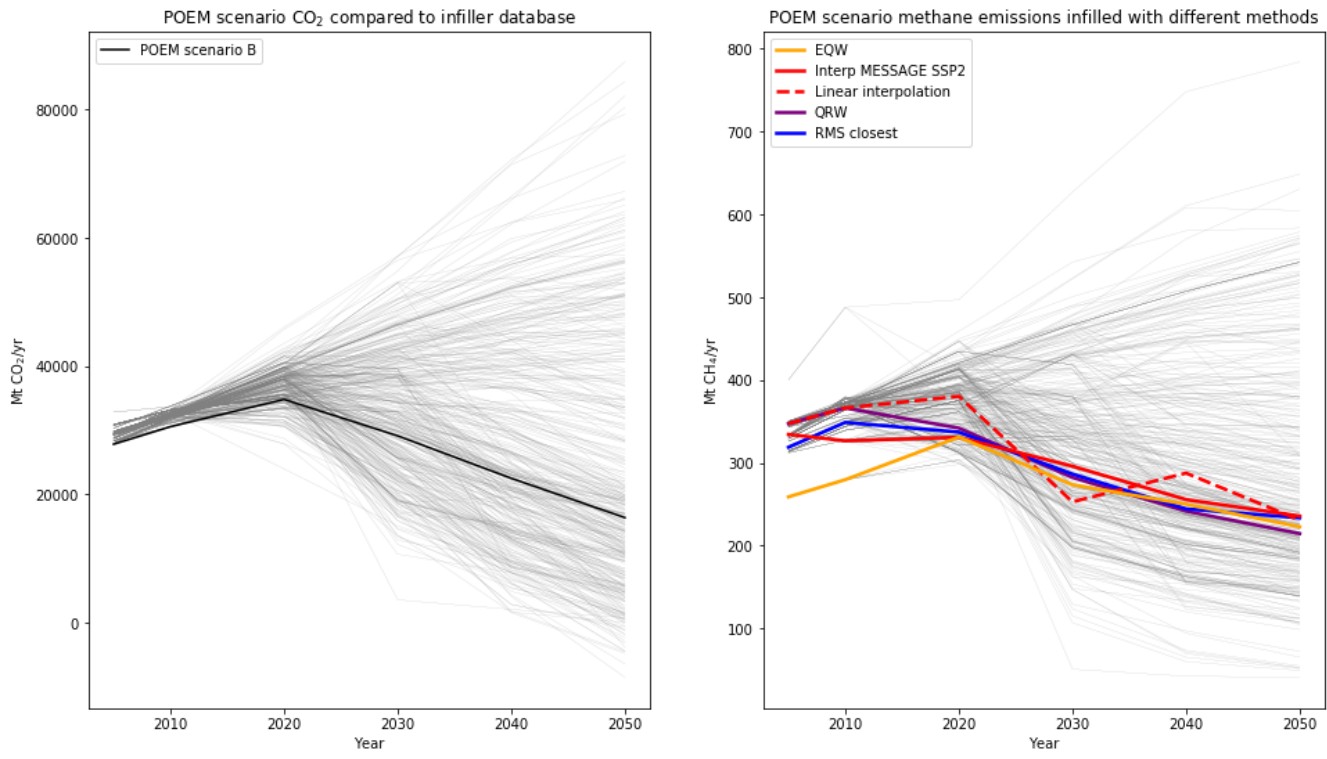

**Figure 4: The Climate Action Tracker (CAT) Kyoto gas totals (thick lines) compared with the portfolio of values in the SR1.5 database (thin lines).**

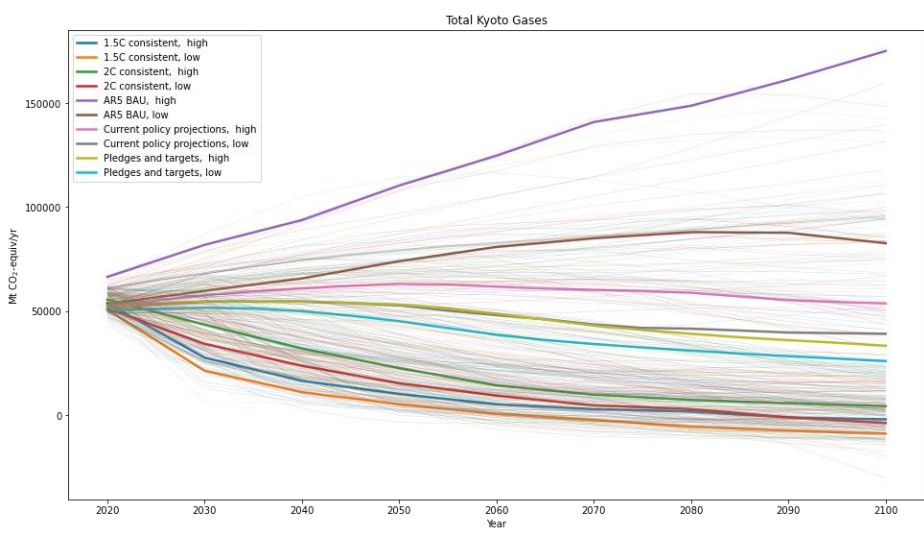

**Figure 5: The CAT Kyoto gas baskets decomposed into their components, using the decompose collection multiple infiller.**

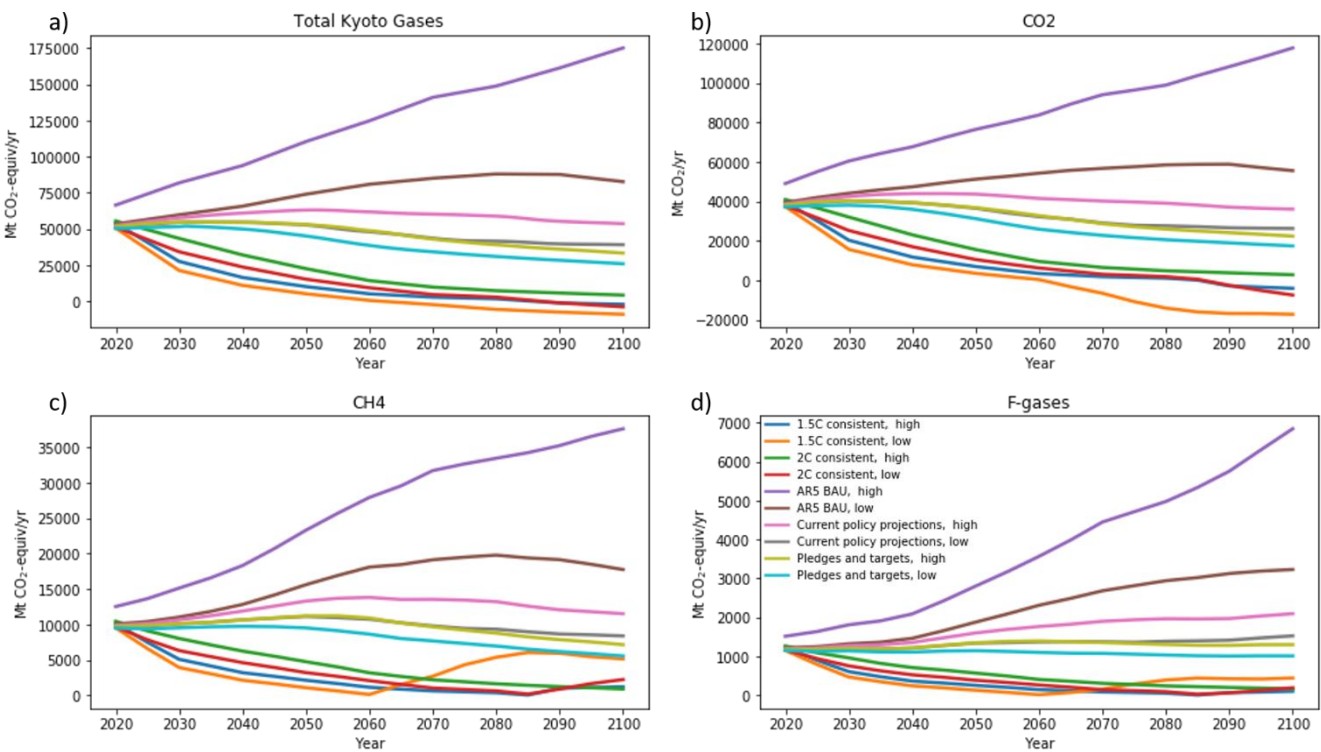

**Figure 6: Kyoto gases, decomposed by first infilling the non-negative emissions using the (non-ratio) quantile rolling windows, then infilling the $CO_2$ using infill composite values.**

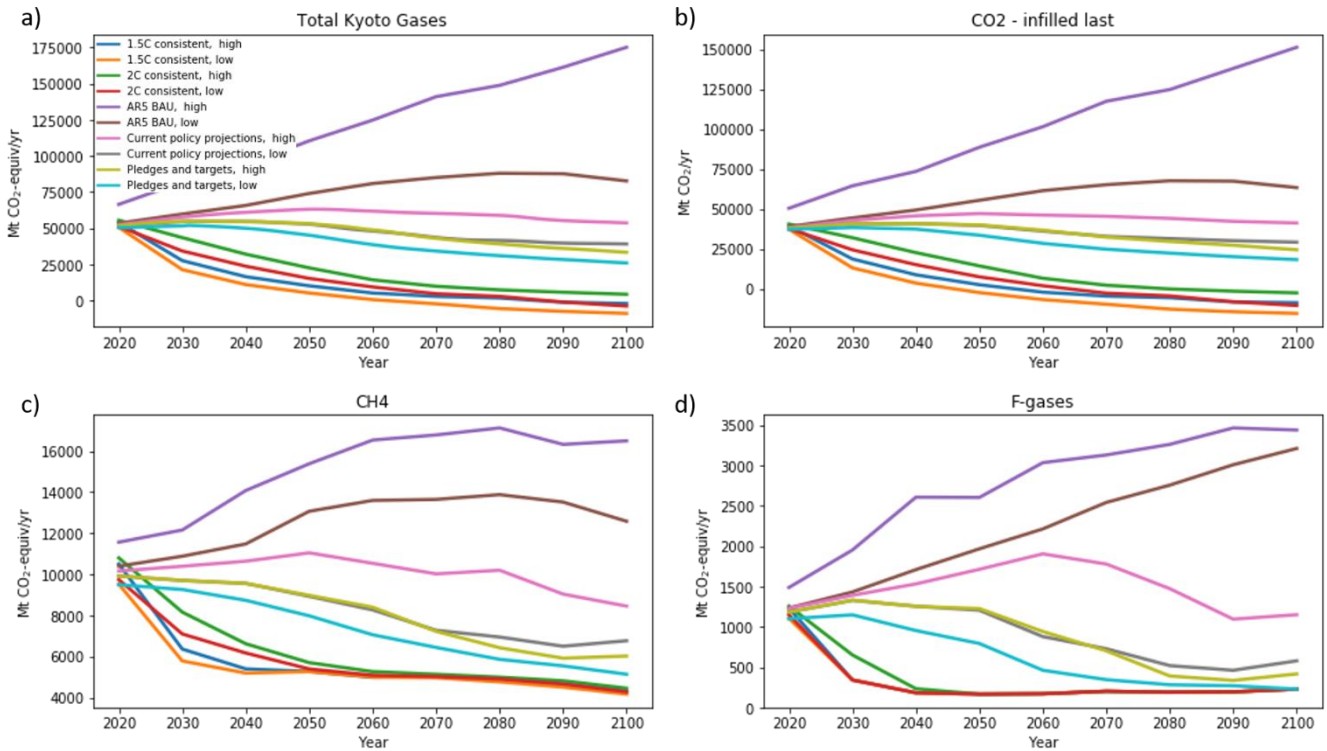

**Figure 7: Illustration of using the interpolate specified scenario cruncher to infill a series of stylised trajectories (solid lines), characterised by two different parameters ($\tau$ and $E_\infty$), defined in** (Sanderson et al., 2016)**. The first column compares the total CO$_2$ calculated for the stylised trajectories to the values of the MESSAGE model for a given group of SSP scenarios (dotted lines). These are our lead values in each case. The second column shows the range of follow values for that SSP. The third column shows the resultant AFOLU (Agriculture, Forestry and Other Land Use) trajectories that emerge from using the Interpolate Specified Scenario infiller.**

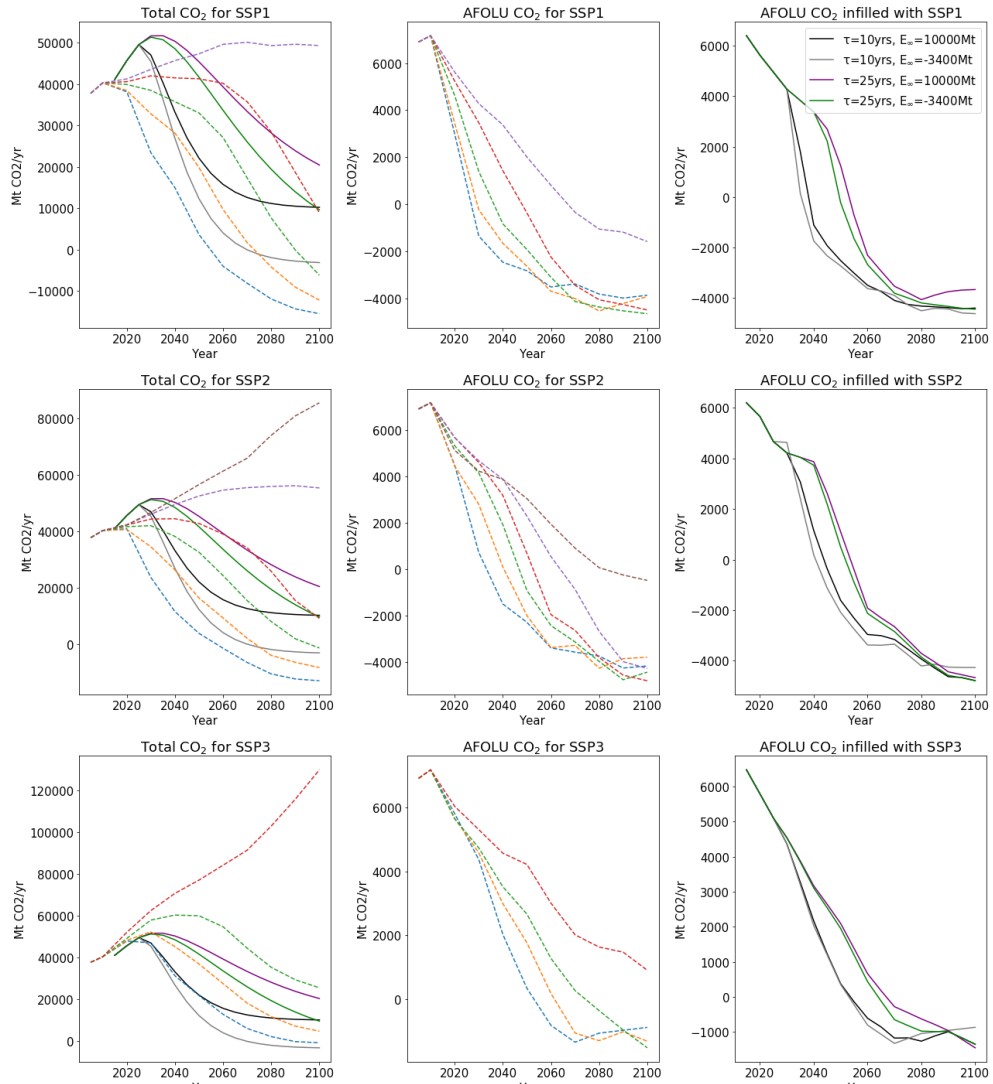

