# Peer review of "Silicone v1.0.0: an open-source Python package for inferring missing emissions data for climate change research"

_Geoscientific Model Development, 2020_

## Referee Comment (RC1) · Anonymous Referee #1 · 24 Jun 2020

**1   General assessment**

The methods and software package described in the manuscript will prove useful for the IAM and climate policy analysis community and are definitely worth publishing. The IAM community will benefit from the easy to use and flexible methods to fill missing data in their output. The climate policy analysis community will benefit as well as incomplete data in scenario databases can hamper analyses and influence the results. Filtering for complete scenarios excludes some models and studies from the analyses while creating your own methodology to fill missing data is time consuming and takes away time from the actual analysis.

However, the manuscript needs some work as some aspects are hard to understand and the structure of the manuscript should be improved.

It is very good that usage examples that actually reproduce the paper's results are given including code, however, the jupyter notebook contain some problems and needed fixes (at least on my computer).

Furthermore, there is still room for improvement of the methods especially for splitting gas basket pathways to individual gases.

I have some general comments followed by more detailed comments. I don't ask the authors to implement all of those comments, most are actually just questions and ideas and I think some answers might be interesting to more readers and could be answered in the manuscript.

**2   General comments**

- It could be helpful for users to briefly mention the package infrastructure used by silicone and if and how it is connected to the other packages in the IAMC toolbox (e.g. data transfer using a common csv format, or by API calls to other packages)

- For me an overview over the processing steps would be great. This would also help to identify possible further use cases and the options to expand the gas basket splitting functionality such that it can replace the EQW (see also comment below.) (see also comment on line 83)

- It would be good to have more structure in explaining the different crunchers and infilliers (e.g. paragraph headings for each cruncher) The paper will also be used as a reference for silicone and then it's good to easily find information on a specific issue. The information on crunchers and infillers is somehow scattered through the document with bits in the methods section (text and tables) and some aspects

**[GMDD](GMDD)**
in the results section. The overview tables are a good idea, but they should be placed in the methods section for the print version of the paper.

- It is a bit unclear how robust the results are and how much work is needed to check them before using the results. It is mentioned in some places, especially the tables, that incomplete databases can influence results. For easy use of the package it would be great if the user is warned in these cases. As one purpose of the package is to fill those gaps it would be a problem if the results are unknowingly influenced by the gaps. Maybe you could add a short section on limitations which gives an overview over such possible problems and references the tables for details. See also comments on lines 99-103 and lines 130-133.

- The gas basket splitting functionality seems to be not fully developed. The package has the potential to replace the EQW and it's update used by assessments like the Climate Action Tracker, but in it's current state it can not yet do that. It would be great if using QRW or EQW with a KyotoGHG constraint was possible for all gases. Could you add information if/how this can be added e.g. as a multiple infiller or a scaling postprocessing step? This would make the package very useful for the climate policy assessment community.

- RMS closest: please consider referencing the paper "Warming assessment of the bottom-up Paris Agreement emissions pledges" (YR du Pont, M. Meinshausen. Nature Communications, 2018) where a similar method has been used.

- It is very useful that the authors test the correlation between the different variables. The Spearman's rank correlation coefficient does not detect non-monotonic relationships which can be modeled by the Quantile Rolling Window method. Have you tried other methods to detect correlations (e.g. the Hoeffding Dependence Coefficient).

- Did you investigate non-emissions lead variables?

[Figure]

- Have you used the method on other emissions databases (AR5DB, SSPDB, ...). Is this easily possible or does the package need a structure only present in the SR15DB?

- My comments on crunchers and infilling methods are a bit scattered as the comments follow the manuscript which also has information on crunchers and infilling methods in several places.

- Finally a more political comment. Why do you feature a fossil fuel companie's scenario in a climate change paper (Shell sky scenario)? This helps their greenwashing attempts. Please provide the scientific explanation for using this scenario.

**3 Detailed comments**

- Abstract: I think it would be good to mention some of the most important use cases and infilling options in the abstract.

- Line 70: It would be helpful for the reader to have short description of the structure of the paper at the end of the introduction.

- Line 83: Please give an overview over the infilling process at the beginning of the methods overview section or by expanding the introduction of the methods section (i.e. what are the steps in the process).

- Lines 99-103: Does the code have any flags to control behavior in these cases? Does it warn if this occurs or is it up to the user to control the results?

- Line 119: Where do the estimates averaged over come from? Different historical estimates for the same year, or different years as well? Is it possible to use

trends of the last years with data or averages over the last years? If only single last historical data points are used (which is what I understand from table 1) the method is very sensitive to annual fluctuations in emissions data. Thus trending or averaging is needed for robust results that do not drastically change by using one additional year with data that has unusual emissions due to e.g. extreme weather or crises (financial, COVID-19, local crises, ...).

- Line 123: I assume "cases" refers to scenarios and scenarios where the sign for the lead variable does not coincide with the infillee are taken out of the average for both lead and follower variable? Maybe revise language to clarify that. A more general point: it's a nice way to avoid numerical problems at transitions to negative emissions. However, when working on small scenario databases and / or with high temporal resolutions you might run into problems with availability of scenarios that use the same time period for the transition to negative emissions as the infillee scenario. What do you do in those cases? Does the algorithm allow for a fallback option or does it just fail?

- Lines 130-133: As a lot of scenario databases are not fully consistent regarding data completeness (and sometimes sector definitions) it is dangerous to not automatically check for completeness or at least have an optional filter that removes incomplete scenarios.

- Line 138: Does "with all the same times" mean data availability for the same points in time? The use of "times" seems unusual and is not intuitive to understand for me.

- Line 142: Line 140 states that the average is taken over all points in time, yet line 142 states that "the value of the lead at one time impacts the whole timeline". That sounds contradictory to me. Can you explain? Also, adding a formula would help.

[Figure]

- I have quite a lot of comments on the quantile rolling window cruncher because it was not straight forward to fully understand your algorithm. I rephrased parts of the algorithm in my own words to understand it and to check if I understood correctly. Maybe this can also give you hints on possible rephrasing to ease understanding for the reader:

  – If I understand correctly, the windows are not windows with clear boundaries but created by the weighting functions. I think the phrasing "Five windows are drawn" is a bit misleading because to me it suggests that the data-points are binned. However, my understanding is that you actually create a smooth distribution from each discrete data point such that at every lead emissions level chosen to calculate a quantile you have all data follower emissions data-points available, just weighted by the weighting function.

  – Please introduce an index for the data points used for the analysis. E.g. the formula in line 160 appears to use $p$ as the sum index, but I assume it actually sums over $w_p(e_l(i))$ for all scenarios i used at evaluation point p (lead emissions $e_p$).

  – If I understand correctly, you calculate known quantile levels $q_i$ through the summing of the ordered (by ascending $e_{f,i}$) normalized weights. For each of the five lead emission levels you analyze and each point in time, you obtain a quantile level $q_i$ for each follower emissions level $e_{f,i}$ in the scenario database. This creates a lookup table where you can search for a given target quantile (usually 0.5, I assume), obtain the adjacent higher and lower quantiles and use these results to find the suitable follower emissions level by interpolation between the points.

  – As the quantiles are calculated for 5 lead emissions levels per point in time the above is done for two lead emissions level $e_p$ and then linear interpolation is used to obtain the follower emissions for the actual lead emissions level. This leads to a further question: why do you use the approach of in-

terpolating between only 5 points. When calculating the quantiles each time you use the method, you could just calculate them for the exact value of the lead emissions. If you pre-calculate the quantiles and use a lookup table to fill emissions (as done in the old EQW), you could use a higher resolution than just 5 points. I get the impression that you kept some aspects of the Generalized Quantile Walk that are no longer necessary with your improved method, because you don't have the binning.

- Line 149: Maybe add "for each time step" to clarify that the operation is carried out at each time step individually. This information could also be added in the description of the process in the following lines.

- Line 149: If I understand correctly you don't interpolate between quantiles, but between follower emissions associated to a given quantile at given lead emission values. If that's correct, please rephrase the sentence.

- Line 155: There is an additional full stop.

- Line 162: The formula uses $r_i$ as sum index, but neither $r_i$ nor $r_p$ are introduced in the text. (I assume $r_i$ are the lead emission data points in the scenario database ordered by the associated follower value and $r_p$ is the highest one to be taken into account at quantile $q$ (so should it not be $r_q$ instead of $r_p$?))

- Line 181: Why did you choose 0.03?

- Line 215: Did you analyze if the RMS closest cruncher chooses scenarios coming from the same model or study such that chosen pathways are very similar?

- Line 252: The sentence starting with "that" is incomplete.

- Line 256: Does the EQW use interpolation between quantiles and smoothing before calculation of the quantiles?

- Lines 259 - 262: Did you test the influence of the weighting function on the conservativeness of the QRW cruncher. I assume that in the extreme case of constant weighting functions this would always return the median of the scenario database. So the smoothing through the weighting function has to be used carefully.

- Line 275: See also comment on line 123. How many scenarios are left when restricting to scenarios with same sign. This will especially be problematic for extreme stylized scenarios with early negative emissions.

- Line 279: a similar approach has been taken in "Warming assessment of the bottom-up Paris Agreement emissions pledges" (YR du Pont, M. Meinshausen. Nature Communications, 2018)

**4  Comments on testing the examples repository**

- There are problems with directory names. The input data directory in the repository is called "input", while in the code is is referenced as "Input" with upper case "I". This leads to the code failing on unix systems.

- Splitting up basket: The notebook fails in cell 7,line 8 with: "TypeError: convert_unit() missing 1 required positional argument: 'to'""

- Stylized Path: the legend of the plot resulting from cell 9 shows 4 cases of MESSAGE scenarios, yet in the plot only two can be distinguished. Is that correct?

- Is it possible to store calculated quantiles for the QRW and EQW method to increase calculation speed by using a lookup table? itemize

**5  Comments on tables and figures**

- Table1: latest time ratio: see comment on line 119. Please explain what re-harmonizing means in this context.
- Table1: Time dependent ratio: when using the optional filtering for same sign, how do you ensure that there is a sufficient number of scenarios available? See also comment on line 123.
- Table 1: Linear interpolation: Is it possible to use scenario binning here to not select two very similar pathways (e.g. same model and storyline with small parameter variation)? I think this could be made more stable by using more than two scenarios, but the problem will not vanish. Can you give an example, where using this cruncher makes sense?
- Table1: EQW: Does the EQW cruncher use smoothing through a weighting function when calculating the quantiles?
- Tables 1 and 2: please repeat the table header after a page break
- Fig 2b: there are some artifacts around the subfigures (partly visible plot boundaries?)
- Fig 3: The spike in the linear interpolation pathway shows that this cruncher is very problematic.
- Figs 4-6: please add a legend

[Figure]

---

## Referee Comment (RC2) · Anonymous Referee #2 · 2 Jul 2020

The paper by Lamboll et al. describes an open-source python package, Silicone, which comprises a collection of algorithms to derive emission pathways of gasses missing from a certain data source using the pathway of an available gas (lead gas) together with the pathway of the available and the missing gas from other data sources. The core of the paper is the description of the currently implemented algorithms (termed crunchers) together with a guide for when to apply which of these crunchers. In addition, the authors try to derive a recommendation for which gas to use as a default lead gas and they demonstrate the applicability of the tool on different test/use cases.

[Figure]

Overall, I find the paper in large parts fluent and well-structured and, albeit I find this collection of algorithms a rather small scientific contribution in itself, it seems to be a self-contained part of a larger pipeline for climate assessments and therefore probably a valuable contribution for climate research within the scope of GMD.

**General comments**

1. In most parts of the manuscript and in the source code the specified aim is to complete missing emissions in future scenarios produced by IAMs.

a) However, in some sections of the manuscript, other aims are indicated (e.g. complete stylised scenarios, fill missing sectoral data, use historical estimates, aggregate regional data(?)) which I found confusing sometimes. One solution could be to remove these hints of other applications and use the (IAM/stylized) future scenarios as the aim and example throughout the paper. Additional applications could then be outlined in a discussions section (in a bit more detail).

b) With respect to the main aim I would have expected use cases showing the completion of several variables of different IAM scenarios.

2. I would recommend to rework parts of the abstract and of the introduction in order to better cover the content/ set the scene for the rest of the manuscript (e.g. add info about rank correlation, tests/ use cases).

3. Literature: it is rather difficult for me to imagine that there are no other somewhat comparable tools around and that so far missing emissions were usually set to zero or only somehow unsystematically filled following 'ad hoc' decisions (as stated in the introduction). For completeness it would be nice if the authors could dig some more into the literature and check how climate models so far got the required input from IAMs? One example for a tool covering a similar purpose in maybe a slightly different but connected setting is the tool used in Gütschow et al. (2016) and du Pont et al.

[Figure]

(2016) which is described in Nabel et al. (2011). Some of the co-authors have been involved in these papers.

4. While reading I sometimes got confused by different terms and I found parts of the manuscript a bit sketchy or difficult to read. More specific:

a) There are several changes in terminology among different (sub-) sections (e.g. "lead variable", "lead gas" "inputs or outputs" (l.89), and timeseries; and a sudden switch to model and scenario in 2.2.2 where 2.2.1 only had the more general term database; but also small things as the change from CH4 in section 1 and 2 to methane starting section 3). I think for the reader it would be helpful to stick to a certain terminology throughout the manuscript.

b) The terms infiller and infilly are very difficult to distinguish in quick reading and I think it would help a lot when choosing less similar terms – How about source and sink/target, or infiller and target, or infiller and silicon-filled, or comparable.

c) In subsection 2.2.2, a bit out of the sudden, several scenarios and models seem to be presupposed, while at the beginning of section 2 only "a database that contains data for at least two emission species" is kind of officially introduced. Maybe it would help to directly introduce the use of different models/IAMs and their scenarios at the beginning of section 2 such that the usage of different timeseries (2.2) and different models and scenarios (2.2.2) is less out of the sudden. An alternative could be a consistent use of the more general terms of "timeseries of different lead variables in the infiller database" depending on the main aim of the paper (see also point 1).

d) Please consider to better structure 2.2.1 and 2.2.2 (e.g. print algorithm names in bold or in italics, with separated paragraphs for the different algorithms or /and as lists (e.g. latex 'description' or the like)).

e) Equations are throughout embedded in the text (see also point 5).

5. Since the different algorithms for the completion of emission timeseries are the

main scientific contribution of the paper I would appreciate if the equations could be clearly separated from the text (i.e. introduced as separated numbered equations) and, furthermore, if more equations would be added (see also specific comments below). In my opinion this could increase readability (4e) and reproducibility. Ideally it could also help to better understand how cases of several lead variable pathways are treated in the different algorithms.

6. Test and use cases only show the usage of absolute value based algorithms, which I find unfortunate and a bit incomplete given the share of the method section dedicated to the ratio based algorithms. How about at least including examples using the time dependent ratio method?

7. Please check the format of your references in the text (e.g. l.28, l.59, l.60, l.68, ...)

**Specific comments/questions**

l.1 Why Silicone?

l.16 Transition. E.g. "In this paper..."

l.16 Please consider to add more information here about the content of the paper

l.33 ... exert ... between? Please check language

l.36 Is there an example reference/study where filling with zeros has been applied?

l.41 I do not understand "does not scale easily"

l.45 There is no 3.8.1 in this reference (reference currently points to Chapter 4, if you target 3.8.1 consider citing Teske et al.)

l.60 Please specify what "this" refers to

l.64 "suite of tools" are these all python tools?

l.66 Are there more than these two? Else consider to add "and/as well as" before

"harmonise..."

l.67 Consider deleting ", managed by the OpenSCM community"

l.82 several? Three/two. . .

l.85 Maybe change to "Currently, there are ..."?

l.98 Consider rephrasing e.g. "... and where emissions are expected to scale with each other ..."

l.98 What do you mean with "regional data" and "aggregate data"? If you refer to regions as subset of global data then this is the first time that a spatial reference is given and I wonder if it would be appropriate to introduce this more formally earlier in the manuscript?

l.100 Consider to give an example? CO2 uptake?

l.106 What do you mean with similar – similar magnitudes?

l.111 "estimate the ratios" – if not to be predefined. . .

l.112 "follower value in infillee database" -> "in the ..."

l.112 Please consider to visually separate (and number) the equations.

l.114 "each different timeseries" -> different regarding what - do you mean different follower variables?

l.116 mean regarding what - time or different sources (models, scenarios?)

l.117 what does "both" refer to?

l.119 what do you mean with "all estimates"? Different sources (models, scenarios)?

l.120 why historical? Couldn't this also be different scenarios from different IAMs?

l.120 "and the lower case $e_f(t)$ represents the follower values in the database at time

t." -> "and the lower case $e_f(t_{last})$ represents the follower values in the database."?

l.125 the infillee lead is not part of the formula – is this the $E_l(t)$ from l.113?

l. 125-133: Maybe consider to restructure? You could start with the context, i.e. the algorithm name, and then add the explanation, e.g.: "The decompose collection multiple infiller is based on... relying on the useful property of ..."

l.134-137: Equation for R(t)?

l.142 I did not understand this sentence

l.143-145: Equation for $E_f$?

l.151 There is no 3.8.1 in that reference (reference currently points to Chapter 4, if you target 3.8.1 consider citing Teske et al.)

l.152 There seems to be a lost copy of the figure caption in the text. (Either just delete or maybe rephrase to steps with complete sentences)

l.155-165 Equation for $E_f$?

l.163 Maybe $E_f$ not $E_l$?

l.166 What about the KyotoGHGs as one basket?

l.169 Maybe give an example for two such variables?

l.185 Consider explicitly listing the two constituents

l.186 Which are "these two" – CO2 and CH4?!

l.188 you write that BC, CO and OC "correlate poorly with others, however, from the table it seems that they do not correlate less well with others than other gasses, the main difference is that they correlate very well with each other.

l.190 maybe aggregate F-gas emissions / F-gasses as a basket?

l.191 up to here always "CH4"

l.198 consider deleting "and find similar results"

l.199 "we choose four" – which are basically all?

l.199 But even if there are errors, wouldn't it be interesting to see what happens? I would appreciate if you could also show results for the 'time dependent ratio' algorithm

l.201-202 Equation?

l.203 "both cases" – CO2 and CH4?

l.203 "non-CO2 pathways" – but CO2 is derived with CH4 -> maybe replace by "emission pathways"

l.204 I would rephrase this, if QRW would be fairly similar all four would be?

l.208 What do you conclude from the non-Gaussian distribution test?

l.212 "either of CO2 or CH4"

l.214 capital T for Table 3

l.225 Add Silicon -> "Data in the Silicon package"

l.238 treatment of regions has not been introduced, maybe explain better or consider deletion?

l.243 "this database" – which? The SR1.5 repository?

l.249-252: Again a lost copy of a figure caption

l.285-287 "free variables" are mentioned twice but are not further explained?

l.287-288 I do not understand this sentence

l.300 consider deleting "of which there are many" or maybe replace by "several options" or the like

Table2: Please explain the asterisk again in this figure caption

Table3: Consider to change the colouring – to me the yellow/orange highlighting gives a 'positive' impression. Maybe you could colour the cells with bold numbers in green and those which are currently yellow in red?

**References**

du Pont, Y.R., Jeffery, M. L., Gütschow, J., Christoff, P. Meinshausen, M. National contributions for decarbonizing the world economy in line with the G7 agreement. Environ. Res. Lett. 11, 054005, 2016.

Gütschow, J., Jeffery, M. L., Gieseke, R., Gebel, R., Stevens, D., Krapp, M., and Rocha, M.: The PRIMAP-hist national 1208 historical emissions time series, Earth Syst. Sci. Data Discuss., 2016, 1-44, 10.5194/essd-2016-12, 2016.

Nabel, J. E. M. S., Rogelj, J., Chen, C. M., Markmann, K., Gutzmann, D. J., Meinshausen, M.: Decision support for international climate policy–The PRIMAP emission module. Environmental Modelling Software, 26(12), 1419-1433, 2011.

---

## Short Comment (SC1) · 9 Jul 2020

Many thanks for your detailed review. You raise many excellent points, which we will update on shortly. Some comment and explanations:

- Statistics and monotonicity: We only tried measurement techniques that would also ignore nonmonotonic relationships. However the spread of the data is very high, so while it is possible that some relationships might have non-monotonic trends at some times, it's unlikely that we would be confident in finding these. We have investigated by eye all combinations of $CO_2$ and $CH_4$ with every other emission for several different years and see no sign of nonmonotonic relationships. My understanding of the Hoeffding Dependence Coefficient is that it's very computationally expensive to run (particularly since I don't see any examples of runtime-optimised code for it in Python), and doing so on so many noisy combinations of emissions and time is essentially p-hacking.

- The use of the Sky scenario opened up analysis of whether the scenario is really Paris-compliant when other types of emission are included. However the answer is somewhat marginal and doing this rigorously would require a much larger amount of space than a simple example warrants (plus the parts of the pipeline not mentioned in this paper), so we will replace it with another scenario.

- The placement and format of the tables is due to the journal submission format - we hope that they would better place it and sort out headings/page splits properly in the published version.

---

## Short Comment (SC2) · 9 Jul 2020

On General Comment 3, I don't think an EQW-like tool was used in Gütschow (2016):

Gütschow, J., Jeffery, M. L., Gieseke, R., Gebel, R., Stevens, D., Krapp, M., & Rocha, M. (2016). The PRIMAP-hist national historical emissions time series. doi:10.5194/essd-2016-12

Maybe Gütschow (2018) was meant?

Gütschow, J., Jeffery, M. L., Schaeffer, M., & Hare, B. (2018). Extending Near-Term Emissions Scenarios to Assess Warming Implications of Paris Agreement NDCs.

[Figure]

Earth's Future, 6(9), 1242–1259. doi:10.1002/2017ef000781

(Both studies use the framework described in Nabel (2011) though.)

---

## Short Comment (SC3) · 9 Jul 2020

Fantastic effort to provide a fully re-usable open-source implementation of data infilling for climate assessments!

Some minor comments you might want to consider:

l. 207 Maybe it could be worded more clearly what is compared with which aim instead of just "CO2 nor methane ..."?

l. 264 The reference to PRIMAP-crf here is not clear to me. On https://climateactiontracker.org/methodology/historical-data/ it is described as one

of the sources of historical input data for the CAT. Maybe a link to the CAT methodology (or the CAT website in general) would be a good reference (https://climateactiontracker.org/methodology/).

General comment: It is hinted at in the paper but maybe it could be made more clearly that any approach based on relation to other available scenarios or ensembles is bound by the available scenarios which are not part of a distribution of likely futures but rather shaped by intercomparison exercises (e.g. certain technologies, or climate policy goals). In any case, excellent that Silicone provides the possibility to filter and select input scenarios and mentions these considerations.

---

## Short Comment (SC4) · 21 Jul 2020

Thanks for your helpful comments Robert!

1. l. 207 Maybe it could be worded more clearly what is compared with which aim instead of just "CO2 nor methane ..."?

   - We have significantly expanded this section as follows: To determine the appropriate statistics to apply on the errors, we first perform a Shapiro-Wilks test to detect any non-Gaussian aspect for the error distribution. This indicated that neither CO2 nor CH4 are statistically significantly non-Gaussian,

either when analysed separately for each cruncher or as an aggregate.

2. l. 264 The reference to PRIMAP-crf here is not clear to me. On https://climateactiontracker.org/methodology/historical-data/ it is described as one of the sources of historical input data for the CAT. Maybe a link to the CAT methodology (or the CAT website in general) would be a good reference (https://climateactiontracker.org/methodology/)

   - Yes, we cited it as being a significant data source. We note that the paper prefers not to cite grey literature, but will probably make an exception here, so the CAT website is now cited as well.

3. General comment: It is hinted at in the paper but maybe it could be made more clearly that any approach based on relation to other available scenarios or ensembles is bound by the available scenarios which are not part of a distribution of likely futures but rather shaped by intercomparison exercises (e.g. certain technologies, or climate policy goals). In any case, excellent that Silicone provides the possibility to filter and select input scenarios and mentions these considerations.

   - This is very true. We have dedicated a lot of space in the notebooks to it, but not really foregrounded it in the paper. We have added a short passage at the beginning of the methods section as follows: "In all cases, the infillers will perform best if the target data comes from a scenario that is socioeconomically similar to scenarios found in the infiller database. The performance of most crunchers can be improved by filtering out scenarios that are known to assume radically different characteristics like population number before infilling, provided that comparable emissions statistics can be found in the remaining database."

Best wishes

Robin

---

## Short Comment (SC5) · 21 Jul 2020

Many thanks for your useful comments, which we will address in full shortly. One point that we don't fully understand, however, was what you meant by "With respect to the main aim I would have expected use cases showing the completion of several variables of different IAM scenarios."

It is not clear what you expect to see that is different from the "use cases" we provide - in the first use case, one scenario (a different scenario in the forthcoming draft) has many variables completed, in the second and third cases complete variables from multiple scenarios. What exactly is it that you wish were different?

[Figure]

To continue the point made by Robert Gieseke above, we are of course happy to cite more papers (particularly our own work!), although some of these specific examples don't seem to qualify - we will cite the Nabel paper and a different Robiou Du Pont paper (Warming assessment of the bottom-up Paris Agreement emissions pledges), as well as the paper suggested by Gieseke unless there was a strong reason for suggesting other ones.

Updated paper to follow!

Many thanks

Robin
* * *

---

## Author Response (AR1)

**Changes in response to reviewer 1:**

**Response to overview comments**

- It could be helpful for users to briefly mention the package infrastructure used by silicone and if and how it is connected to the other packages in the IAMC toolbox (e.g. data transfer using a common csv format, or by API calls to other packages)

  - We have expanded the section discussing the pipeline to mention data transfer. The section now reads:
    - The pipeline is based around the pyam package (Gidden and Huppmann, 2019), specifically its IamDataFrame class, which Silicone makes extensive use of. Pyam dataframes easily convert from and to widely-used pandas dataframes, which pyam and Silicone also use internally (McKinney, 2011). The pipeline also includes tools to harmonise (i.e., correct projection made in the past to match now-known emissions) (aneris, (Gidden et al., 2018a) before infilling and to pass the complete projections to climate simulators. The estimation of climatic impact is performed by OpenSCM,

- For me an overview over the processing steps would be great.

  - We describe all the processing steps in the Methods section. Other than getting the data into pyam form and harmonising it (if desired) there aren't any other pre-processing steps needed. We've added a description describing the interaction between silicone and the harmonisation process in the mathematical detail section. Is there something else you would like to see discussed?
    - If the results are to be harmonised, then harmonising both the infiller and target data before infilling is required for improved consistency (otherwise infilling depends on outdated data). Absolute value infilling techniques preserve harmonisation, however ratio-based approaches do not necessarily, and may need harmonisation again afterwards.

- It would be good to have more structure in explaining the different crunchers and infilliers (e.g. paragraph headings for each cruncher) The paper will also be used as a reference for silicone and then it's good to easily find information on a specific issue. The information on crunchers and infillers is somehow scattered through the document with bits in the methods section (text and tables) and some aspects in the results section.
    - [Sub-subheadings have been added to the sections detailing the crunchers]
    - [The rank correlations analysis has been added to the results section. Although not formally a result of silicone, this hopefully provides a clearer separation of model description and data.]

- The overview tables are a good idea, but they should be placed in the methods section for the print version of the paper.

  - We agree, but have no control over typesetting – tables were requested to be at the end of the document at this stage. Hopefully the editors will take note!

- It is a bit unclear how robust the results are and how much work is needed to check them before using the results. It is mentioned in some places, especially the tables, that incomplete databases can influence results. For easy use of the package it would be great if the user is warned in these cases. As one purpose of the package is to fill those gaps it would be a problem if the results are unknowingly influenced by the gaps. Maybe you could add a short section on limitations which gives an overview over such possible problems and references the tables for details. See also comments on lines 99-103 and lines 130-133.

  - The code has now been updated so that the default behaviour is to ignore inconsistent data (e.g. scenarios missing values at one time). There is also now a warning when infilling negative lead emissions with the time-dependent ratio.

- - [The text for "time-dependent ratio" and "decompose collection with time-dependent" ratio has been changed significantly to reflect this.] "This relies on all scenarios having values for all of these variables, so misses out cases which do not have one of the constituents or only reports at some of the required times, unless the override option "only_consistent_cases" is set to False."
- The gas basket splitting functionality seems to be not fully developed. The package has the potential to replace the EQW and it's update used by assessments like the Climate Action Tracker, but in it's current state it can not yet do that. It would be great if using QRW or EQW with a KyotoGHG constraint was possible for all gases. Could you add information if/how this can be added e.g. as a multiple infiller or a scaling postprocessing step? This would make the package very useful for the climate policy assessment community.
  - Since writing the first draft we made another multiple infiller that does exactly this. We have now included details of it in the paper. Note that one of the gases is infilled by the conservation condition
    - ["Split collection with remainder emissions" entry added to multiple infiller table and flow chart, described the process of infilling differently for breaking up the aggregate values into their components]
- RMS closest: please consider referencing the paper "Warming assessment of the bottom-up Paris Agreement emissions pledges" (YR du Pont, M. Meinshausen. Nature Communications, 2018) where a similar method has been used
  - Good suggestion. Included as follows:
    - The alternative approach of inputting the whole pathway with the smallest mean-squared distance over all time was used (Robiou du Pont & Meinshausen, 2018). This works well for large databases containing similar paths, but is less useful for smaller databases or for paths with an unusual behaviour over time.
- It is very useful that the authors test the correlation between the different variables. The Spearman's rank correlation coefficient does not detect nonmonotonic relationships which can be modeled by the Quantile Rolling Window method. Have you tried other methods to detect correlations (e.g. the Hoeffding Dependence Coefficient).
  - As mentioned in the previous correspondence, we have tried only methods that would also not detect non-monotonic trends. Correlations for all variables with CH4 and CO2 were also plotted and inspected by eye for three times and no clear nonmonotonicity was seen, however.
    - We also plotted the relationships between $CO_2$ and all other variables (using the plotting function in the Silicone examples github) to check that there were no obvious cases of a non-monotonic relationship.
- • Did you investigate non-emissions lead variables?
  - No, although the mathematics should work identically. We now point out using economic indicators as a potentially use-case for ratio infillers, although strictly speaking it's not "infilling" if this data is used (it's just a very basic IAM).
- Have you used the method on other emissions databases (AR5DB, SSPDB, ...). Is this easily possible or does the package need a structure only present in the SR15DB?
  - As described below, we now apply Silicone techniques to infill a model/scenario from AR5. It is also being applied to AR6. Given that all these listed databases use the IAMC format (more or less), they are easily handled by pyam and hence Silicone. The only pinch point for using other databases is getting the data in the pyam structure. Many databases are in pyam or pyam-compliant structures, so following on from the point below we now show infilling a scenario from such a source.
- Finally a more political comment. Why do you feature a fossil fuel companie's scenario in a climate change paper (Shell sky scenario)? This helps their greenwashing attempts. Please provide the scientific explanation for using this scenario.
  - This was originally referenced because that scenario is claimed to be Paris-compliant and was also one the industry scenarios included in the IPCC SR1.5 pathway assessment. However, it is our contention that this

90    depends on the way that the scenario is infilled. However since to demonstrate this point would require careful data handling, harmonisation and running of a climate simulator to show convincingly we will leave that discussion for elsewhere.

- [The Sky model infilling has been replaced by POEM scenario B infilling from AR5. Text and images have been updated to show this.]

95

**Detailed comments – note that line counts are drastically different now**

- Abstract: I think it would be good to mention some of the most important use cases and infilling options in the abstract.
  - Good idea, we have added a sentence
100      - We demonstrate the package's utility with three examples: infilling all required gases for a pathway with data for only one emission species, splitting up a Kyoto emissions total into separate gases and complementing a set of idealised emissions curves to provide a complete, consistent emissions portfolio.
- Line 70: It would be helpful for the reader to have short description of the structure of the paper at the end of the
105  introduction.
  - Good idea
      - This paper is structured as follows: the *Methods* section presents an overview of the different infiller methods, then goes through the infiller techniques in precise and mathematical detail. In *Results*, we present our analysis of emissions projections in the SR1.5 database. This includes
110        correlation statistics of the database, and how well Silicone reproduces aspects of it from the rest. We use this to draw conclusions on the implications for using Silicone on unknown data. In *Use Cases*, we present three examples of using Silicone for infilling a pathway with limited information, splitting up an aggregate basket of emissions and infilling stylised emissions trajectories. We end with a summary of our paper.
115  - • Line 83: Please give an overview over the infilling process at the beginning of the methods overview section or by expanding the introduction of the methods section (i.e. what are the steps in the process).
  - This is now included, if you mean this in a philosophical sense. We also now discuss elsewhere the protocol for harmonisation. If you mean in terms of which commands to use in what order, that seems best learnt from code examples, found in the notebooks.
120      - Silicone offers a range of tools that apply methods for doing this infilling which are appropriate in different circumstances, depending on the amount of complete data and how much we know about the narrative behind our emissions. These tools are referred to as 'crunchers'. Each of these crunchers takes a 'lead variable', found in both the infiller and target databases, and uses it to infer the value of a 'follower variable', found only in the infiller database (hence missing in the
125        target database). There are also several tools for easily infilling multiple variables, called 'multiple infillers'. These may have multiple follower or lead variables.
      - If the results are to be harmonised, then harmonising both the infiller and target data before infilling is required for improved consistency (otherwise infilling depends on outdated data). Absolute value infilling techniques preserve harmonisation, however ratio-based approaches do
130        not necessarily, and may need harmonisation again afterwards.
- Lines 99-103: Does the code have any flags to control behavior in these cases? Does it warn if this occurs or is it up to the user to control the results?
  - A warning is now reported for ratio methods with negative target leads, although it is up to the user to determine if the use is acceptable or not.

135     • Line 119: Where do the estimates averaged over come from? Different historical estimates for the same year, or different years as well? Is it possible to use trends of the last years with data or averages over the last years? If only single last historical data points are used (which is what I understand from table 1) the method is very sensitive to annual fluctuations in emissions data. Thus trending or averaging is needed for robust results that do not drastically change by using one additional year with data that has unusual emissions due to e.g. extreme weather or crises

140       (financial, COVID-19, local crises, ...).
         o The estimates are not averaged over any period. The vast majority (in our case all) of projected emissions trajectories data reported by integrated assessment models is available at 5 or 10 year intervals. Such data points therefore aim to represent systemic changes rather than (sub-)annual effects. Still, in the case of emissions that fluctuate strongly over time, this infiller should be regarded as less reliable. In the

145          manuscript, we describe the potential use case of this infiller, and we now have added this pitfall in Table 1.
    • Line 123: I assume "cases" refers to scenarios and scenarios where the sign for the lead variable does not coincide with the infillee are taken out of the average for both lead and follower variable? Maybe revise language to clarify that. A more general point: it's a nice way to avoid numerical problems at transitions to negative emissions.

150       However, when working on small scenario databases and / or with high temporal resolutions you might run into problems with availability of scenarios that use the same time period for the transition to negative emissions as the infillee scenario. What do you do in those cases? Does the algorithm allow for a fallback option or does it just fail?
         o You assume correctly (rephrased for clarity now). In this case, the infiller hard fails. The user should decide explicitly whether to use the sign-independent version or another infiller.

155              ▪ It will produce an error if there is no data with the required sign.
    • Lines 130-133: As a lot of scenario databases are not fully consistent regarding data completeness (and sometimes sector definitions) it is dangerous to not automatically check for completeness or at least have an optional filter that removes incomplete scenarios.
         o This is indeed somewhat dangerous. We have now changed how this works so that by default, data that

160          lacks either some of the constituent variables or only reports at some of the required times is removed before infilling.
             ▪ This cruncher is the foundation for the 'decompose collection with time-dependent ratio' multiple infiller. It relies on all scenarios having values for all of these variables, so misses out cases which do not have one of the constituents or only reports at some of the required times, unless the

165                  override option "only_consistent_cases" is set to False. It always constructs a new, consistent version of the aggregate variable in case different modellers used different conversion factors in the infiller database.
    • Lines 138: Does "with all the same times" mean data availability for the same points in time? The use of "times" seems unusual and is not intuitive to understand for me.

170          o It does, rephrased
             ▪ The 'RMS closest' cruncher filters the infiller database for models with data at all the times found in the infillee data.
    • Line 142: Line 140 states that the average is taken over all points in time, yet line 142 states that "the value of the lead at one time impacts the whole timeline". That sounds contradictory to me. Can you explain? Also, adding a

175       formula would help.
         o Rephrased, formula added
             ▪ The 'RMS closest' cruncher filters the infiller database for models with data at all the times found in the infillee data. It then ranks models and scenarios by the root mean squared (RMS) difference between the lead data in the infiller and infillee database, with the average being taken over all

180              timeslices. It returns the follower data from the scenario/model combination with the smallest RMS difference: the formula is $E_f(t) = e_{f,i}(t)$, where the subscript $i$ refers to the model/scenario

case that minimises $\sum_t \left( E_l(t) - e_{l,i}(t) \right)^2$. In the case of a draw, the value that occurs earlier in the infiller database will be used. This is the only cruncher that is not time-independent, i.e. changing the value of the lead at one time may result in different outputs at other times.

- [Many comments about QRW] If I understand correctly, the windows are not windows with clear boundaries but created by the weighting functions. I think the phrasing "Five windows are drawn" is a bit misleading because to me it suggests that the data-points are binned. However, my understanding is that you actually create a smooth distribution from each discrete data point such that at every lead emissions level chosen to calculate a quantile you have all data follower emissions
  data-points available, just weighted by the weighting function.
  - o Mostly correct, one other possible misconception is that we default to 10 windows, 5 was just chosen for illustrative reasons.
    - ▪ A number of rolling windows centers (here 5, by default 10) are drawn and a weighting function constructed for each window. It has a continuous distribution, rather than a discrete cutoff, hence the name.
- Please introduce an index for the data points used for the analysis.
  - ▪ [We have significantly changed the notation for the formulae to do this.]
- why do you use the approach of in-terpolating between only 5 points. When calculating the quantiles each time you use the method, you could just calculate them for the exact value of the lead emissions. If you pre-calculate the quantiles and use a lookup table to fill emissions (as done in the old EQW), you could use a higher resolution than just 5 points. I get the impression that you kept some aspects of the Generalized Quantile Walk that are no longer necessary with your improved method, because you don't have the binning.
  - o The five points we have used are mainly for plotting clarity. It is up to the user how many points they wish to use and they could use arbitrarily high numbers of points (we default to 10 because, in practice, there is little variation at higher resolution than this). Another reason for using fewer pointsis computational simplicity – calculating a quantile at each point is O(N) in the infiller database, so doing so for each point in the infillee database can get time-consuming for larger infillee databases.
- Line 149: Maybe add "for each time step" to clarify that the operation is carried out at each time step individually. This information could also be added in the description of the process in the following lines.
  - ▪ We have added "for each time."
- Line 149: If I understand correctly you don't interpolate between quantiles, but between follower emissions associated to a given quantile at given lead emission values. If that's correct, please rephrase the sentence.
  - o That was indeed not the interpolation we were trying to talk about here
    - ▪ infills the values based on interpolating between the quantiles of the follower variable.
- Line 155: There is an additional full stop.
  - ▪ [At some points, the whole text of the caption was used rather than just a reference. This has been deleted]
- Line 162: The formula uses ri as sum index, but neither ri nor rp are introduced in the text. (I assume ri are the lead emission data points in the scenario database ordered by the associated follower value and rp is the highest one to be taken into account at quantile q (so should it not be rq instead of rp?))
  - ▪ [Significantly rewritten in accordance with the above]
  - ▪ $q(e_l(j)) = \sum_{e_f(i)<e_f(j)} w_p(e_l(i)) + \frac{w_p(e_l(j))}{2}$.
- Line 181: Why did you choose 0.03?
  - o To select only extreme cases.

- We also calculate the variation of this value with time, and in cases where this exceeds 0.03 (chosen to highlight only extreme cases),
- Line 215: Did you analyze if the RMS closest cruncher chooses scenarios coming from the same model or study such that chosen pathways are very similar?
  - We didn't in detail, however they typically this won't happen as the same model will have a large spread in well-modelled variables between the scenarios by construction (there's little point in repeating yourself). By definition, the RMS closest cruncher will choose similar pathways in terms of trend.
- Line 252: The sentence starting with "that" is incomplete.
  - Again, a figure-caption error
    - [deleted]
- Line 256: Does the EQW use interpolation between quantiles and smoothing before calculation of the quantiles?
  - We have added have a section explaining this properly, as follows
    - The equal quantile walk calculates the quantile of the lead value at each time. This 0 for values below the database minimum, 1 for those above the database maximum and the fraction of infiller data smaller or equal to this value otherwise. We interpolate between neighbouring values in the infiller data to avoid rounding errors.
- Lines 259-262: Did you test the influence of the weighting function on the conservativeness of the QRW cruncher. I assume that in the extreme case of constant weighting functions this would always return the median of the scenario database. So the smoothing through the weighting function has to be used carefully.
  - We did, you are correct.
    - Increasing the decay length will reduce the weight difference between points, so the rolling window becomes wider and more even, with the limit case of calculating quantile $q$ of all data for large $d_l$.
- Line 275: See also comment on line 123. How many scenarios are left when restricting to scenarios with same sign. This will especially be problematic for extreme stylized scenarios with early negative emissions.
  - Indeed, see comments above.
- Line 279: a similar approach has been taken in "Warming assessment of the bottom-up Paris Agreement emissions pledges" (YR du Pont, M. Meinshausen. Nature Communications, 2018)
    - [Reference added]

**Comments on testing the examples repository**

- There are problems with directory names. The input data directory in the repository is called "input", while in the code is is referenced as "Input" with upper case "I". This leads to the code failing on unix systems.
    - [Fixed]
- Splitting up basket: The notebook fails in cell 7,line 8 with: "TypeError: convert_unit() missing 1 required positional argument: 'to'""
  - This problem should not happen with the latest version of silicone, do you still find it?
- Stylized Path: the legend of the plot resulting from cell 9 shows 4 cases of MESSAGE scenarios, yet in the plot only two can be distinguished. Is that correct?
  - Correct, $E_\infty$ does not appear in this formula so two of these lines are identical.

- Is it possible to store calculated quantiles for the QRW and EQW method to increase calculation speed by using a lookup table?
    - The quantiles for QRW are calculated at the point of generating the infiller, so several different target databases can be infilled efficiently in this way, but it doesn't speed up our use case here. More lookup tables are possible in principal but this would not demonstrate the use of this toolkit in practice though. We could add such functionality in future but we don't believe it is necessary to illustrate the tool in this paper (pull requests and issues at the github repository are most welcome). We have reduced the computational load of the new POEM infilling notebook relative to the Shell notebook by removing most of the variables, which were never plotted or explored anyway.

**Comments on tables and figures**
- Table1: latest time ratio: see comment on line 119. Please explain what re-harmonizing means in this context.
    - [Comment has been removed here]
- Table1: Time dependent ratio: when using the optional filtering for same sign, how do you ensure that there is a sufficient number of scenarios available?
    - We raise an error if there are none of the correct sign, It's not clear that there's a "sufficient number" required above 1. There will now be a warning thrown if the target data is negative, irrespective of the number of negative infiller scenarios.
- Table 1: Linear interpolation: Is it possible to use scenario binning here to not select two very similar pathways (e.g. same model and storyline with small parameter variation)? I think this could be made more stable by using more than two scenarios, but the problem will not vanish. Can you give an example, where using this cruncher makes sense?
    - This cruncher is only intended for use on a very limited number of scenarios. We tend to want cases where the scenarios have similar model and storylines. This then ensures that the infilled result is compatible with those storylines. E.g. my model assumes a SSP5 world and is comparable to these groups of scenarios with high biomass burning – I want to infill the amount of BC corresponding to this much $CO_2$. We also now mention that this is very similar to a previously developed interpolator that used cubic spline interpolation, which is fairly similar in the limit of a large database. This specific cruncher in Silicone was recently used in the paper *Current and future global climate impacts resulting from COVID-19* https://www.nature.com/articles/s41558-020-0883-0
        - A tool for infilling was provided with (Rogelj et al., 2014) using a cubic spline between specific points in a small database, however this type of infiller behaves chaotically when applied to large databases incorporating many different models. It was also coded in Excel, limiting the ease of open-source development.
- Table1: EQW: Does the EQW cruncher use smoothing through a weighting function when calculating the quantiles?
    - EQW is smooth but does not have a weighting function
        - [Section on EQW added elsewhere]
- Tables 1 and 2: please repeat the table header after a page break
    - Good suggestion. This would be a matter for typesetters. Tables do not behave properly with track changes so we will not attempt to do this by hand here
- Fig 2b: there are some artifacts around the subfigures (partly visible plot boundaries?)
    - [hopefully fixed now]
- Fig 3: The spike in the linear interpolation pathway shows that this cruncher is very problematic.

- o Correct and intentional. This is partly alleviated in the current version, but we do not recommend using linear interpolation without filtering the data down carefully, as is described in the text.
  - ■ We see from Figure 3 that the linear interpolation model (without filtering the database) provides a chaotic pathway, due to its value being determined only by the two points either side of it in the database, which changes semi-randomly with time and should not be used here.
- Figs 4-6: please add a legend
  - ■ Added legends to the figures.

**Changes in response to reviewer 2:**

**Changes made**

1a. In most parts of the manuscript and in the source code the specified aim is to complete missing emissions in future scenarios produced by IAMs. However, in some sections of the manuscript, other aims are indicated (e.g. complete stylised scenarios, fill missing sectoral data, use historical estimates, aggregate regional data(?)) which I found confusing sometimes. One solution could be to remove these hints of other applications and use the (IAM/stylized) future scenarios as the aim and example throughout the paper. Additional applications could then be outlined in a discussions section (in a bit more detail).

- o To reduce the confusion, we have removed the reference to aggregating regional data entirely and have rephrased the section around historic estimates and sector emissions. Since one of the crunchers (last time ratio) is specifically designed to deal with the case of historic emissions, it is still discussed there. We mention regions in the context of structure as follows: "Pyam dataframes assign values to variables as a function of different models, scenarios, regions and times. All methods work on databases with only a single region at a time, although the region can be different between the infiller and target databases."

1b. With respect to the main aim I would have expected use cases showing the completion of several variables of different IAM scenarios.

- As mentioned in the separate discussion piece submitted previously, we are somewhat confused by this comment – the first "use case" provides a case where a scenario (now a different scenario) has many variables completed, the second and third cases complete variables from multiple scenarios. There is also a github repository consisting only of examples of use (some of which are mentioned in this paper, some not – note that the Sky scenario example in the text has changed now) and a "notebook" section to the main github talking through the general principles of applying these techniques. Could you clarify if there is something else you would like to see added, and whether you really want this included in the paper itself? (We are essentially treating the examples github as a living and evolving Supplementary Info, although the archived version is always available from Zenodo.)

2. I would recommend to rework parts of the abstract and of the introduction in order to better cover the content/ set the scene for the rest of the manuscript (e.g. add info about rank correlation, tests/ use cases).

345        ○  We now mention the use-cases in the abstract and have an outline of the manuscript in the introduction. "A variety of infilling options are outlined and their suitability for different cases are discussed. We recommend certain infilling techniques as the good defaults, but emphasise that considering the specifics of the model being infilled will produce better results. We demonstrate the package's utility with three examples: infilling all required gases for a pathway with data for only one emission species, splitting up a

350    Kyoto emissions total into separate gases and complementing a set of idealised emissions curves to provide a complete, consistent emissions portfolio."

3. Literature: it is rather difficult for me to imagine that there are no other somewhat comparable tools around and that so far missing emissions were usually set to zero or only somehow unsystematically filled following 'ad hoc' decisions (as stated in the introduction). For completeness it would be nice if the authors could dig some more into the literature and check how

355    climate models so far got the required input from IAMs? One example for a tool covering a similar purpose in maybe a slightly different but connected setting is the tool used in Gütschow et al. (2016) and du Pont et al. (2016) which is described in Nabel et al. (2011). Some of the co-authors have been involved in these papers.

    •  As discussed with Robert Gieseke in previous correspondance, some specific papers you mention do not strictly do infilling – they use non-emissions economic data to perform the analysis, whereas we deem 'infilling' to be when

360        no such data is available.
        ○  We have changed the tense of the sentence to clarify that explicitly setting these values to zero is rare, although very often done implicitly by ignoring the variable altogether. In explicit cases, the Equal Quantile Walk is used. We now mention Nabel et al as an example of using this but note that there is no general tool for this so we think our open-source codebase provides a significant step forward in terms of

365            reducing duplicated effort. We also now highlight the use of a RMS-closest technique by Robiou du Pont & Meinshausen, 2018. The text now reads:
        ○  "Most earlier studies overcame this problem in one of two ways: with expert-based ad-hoc decisions on how to adequately fill-in missing species (Schaeffer et al., 2015); or by assuming that a pathway will occur at the same quantile for each set of emissions in a particular year, although the quantile can vary over time

370            (Gütschow et al., 2018; Meinshausen et al., 2006; Nabel et al., 2011). However, the former clearly does not scale easily to larger databases (because making ad-hoc decisions for a thousand scenarios requires a significant time input), and the latter approach, termed the "Equal quantile walk" (EQW), ignores trade-offs and specific relationships between emission species resulting from how competing technologies, behaviours and industrial practices result in different emissions. A few alternative approaches have been

375            used recently: for instance, using the pathway with the smallest mean-squared distance over all time was used in (Robiou du Pont and Meinshausen, 2018). This works well for large databases containing similar paths, but is less reliable for smaller databases or for paths with an unusual behaviour over time. A more sophisticated "Generalized Quantile Walk" technique can capture the effect of trade-offs and was recently introduced in section 3.8.1 in (Teske et al., 2019), involving quantile regression between a lead variable

380            (fossil $CO_2$ emissions) and other gases for every individual year. Unfortunately, the implementation there did not consistently guarantee that higher quantiles resulted in higher emissions, and has not been followed up with any peer-reviewed work that does so. A tool for infilling was provided with (Rogelj et al., 2014) using a cubic spline between specific points in a small database, however this type of infiller behaves chaotically when applied to large databases incorporating many different models. It was also coded in

385            Excel, limiting the ease of open-source development."

4. While reading I sometimes got confused by different terms and I found parts of the manuscript a bit sketchy or difficult to read. More specific:
a) There are several changes in terminology among different (sub-) sections (e.g. "lead variable", "lead gas" "inputs or outputs" (l.89), and timeseries; and a sudden switch to model and scenario in 2.2.2 where 2.2.1 only had the more general term database; but also small things as the change from CH4 in section 1 and 2 to methane starting section 3). I think for the reader it would be helpful to stick to a certain terminology throughout the manuscript.

- References to "lead/follower gas" have been changed to "lead/follower variable". Inputs is now generally removed where it is equivalent to "infiller", see below. Timeseries refers to any data that changes over time and is used to refer to any of the above – it is a data structure, not a description of the data meaning.
  - A short paragraph explaining what model/scenario combinations are has been added before 2.2.1: "As one final detail, we discuss the data model which is assumed by Silicone. Silicone is built around the pyam package (Gidden and Huppmann, 2019). As a result, it assumes that all input data is provided in a particular structure. The structure includes the model which created the timeseries, the scenario with which the timeseries is associated (e.g. a high BECS 1.5 degree scenario), the region the emissions occurs in and the unit of the data (full details available at https://pyam-iamc.readthedocs.io/en/stable/data.html). Accordingly, Silicone is able to work on specific subsets of models (e.g. only the MESSAGE model) or subsets of scenarios (e.g. all SSP1-like scenarios). We therefore follow the pyam convention and refer to a "model/scenario combination" to mean a single projected world, that in some contexts might be called a "scenario"."

b) The terms infiller and infilly are very difficult to distinguish in quick reading and I think it would help a lot when choosing less similar terms – How about source and sink/target, or infiller and target, or infiller and silicon-filled, or comparable.
  - Infiller and target seems like a good combination, now used throughout.

c) In subsection 2.2.2, a bit out of the sudden, several scenarios and models seem to be presupposed, while at the beginning of section 2 only "a database that contains data for at least two emission species" is kind of officially introduced. Maybe it would help to directly introduce the use of different models/IAMs and their scenarios at the beginning of section 2 such that the usage of different timeseries (2.2) and different models and scenarios (2.2.2) is less out of the sudden. An alternative could be a consistent use of the more general terms of "timeseries of different lead variables in the infiller database" depending on the main aim of the paper (see also point 1).
  - Change made as described in 4a), introducing the structure of pyam dataframes.

d) Please consider to better structure 2.2.1 and 2.2.2 (e.g. print algorithm names in bold or in italics, with separated paragraphs for the different algorithms or /and as lists (e.g. latex 'description' or the like)).
  - Sub-subtitles now used, in bold.

e) Equations are throughout embedded in the text (see also point 5).
  - Key equations now in "display" mode.

5. Since the different algorithms for the completion of emission timeseries are the main scientific contribution of the paper I would appreciate if the equations could be clearly separated from the text (i.e. introduced as separated numbered equations) and, furthermore, if more equations would be added (see also specific comments below). In

430  my opinion this could increase readability (4e) and reproducibility. Ideally it could also help to better understand how cases
of several lead variable pathways are treated in the different algorithms.
   o  As above. Important equations are numbered for external reference and we have added several
      equations for clarity.
6. Test and use cases only show the usage of absolute value based algorithms, which
435  I find unfortunate and a bit incomplete given the share of the method section dedicated
to the ratio based algorithms. How about at least including examples using the time
dependent ratio method?
   o  One of the ways of splitting the Kyoto gas totals is the "decompose collection with time-dependent ratio",
      which is a wrapper calling the ratio-based method several times. We have modified the text to make this
440  clearer that this such an example.
7. Please check the format of your references in the text (e.g. l.28, l.59, l.60, l.68, : : : )
   o  This has been corrected.

445  **Specific comments/questions**
l.1 Why Silicone?
   • Our package fills what some would call gaps in emissions scenarios. Silicone is a caulking agent used to fill in gaps
      in tiling and was the first thing one of the developers thought of when we searched for a name. We are happy to add
      an explanation in the text if you feel it is necessary.
450  l.16 Transition. E.g. "In this paper: : :"
   o  "This paper presents a variety of infilling options and outlines their suitability for different cases are
      discussed. We recommend certain infilling techniques as the good defaults, but emphasise that considering
      the specifics of the model being infilled will produce better results. We demonstrate the package's utility
      with three examples: infilling all required gases for a pathway with data for only one emission species,
455  splitting up a Kyoto emissions total into separate gases and complementing a set of idealised emissions
      curves to provide a complete, consistent emissions portfolio."
l.16 Please consider to add more information here about the content of the paper
   • Detail added, as above
l.33 : : : exert : : : between? Please check language
460  o  Changed to "as a large number of supposedly minor emissions may collectively exert a significant
      radiative forcing."
l.36 Is there an example reference/study where filling with zeros has been applied?
   • Implicitly, this is done by every study where any of the F-gases are ignored (e.g. every IAM in the SR1.5 database
      ignores $NF_3$). However, studies don't tend to list the gases they ignore.
465  o  "If no infilling is attempted, the unevaluated emissions would effectively be considered zero, which would
      clearly create systematic biases and potential artefacts in the projected temperatures."
l.41 I do not understand "does not scale easily"
   o  Added "However, the former clearly does not scale easily to larger databases (because making ad-hoc
      decisions for a thousand scenarios requires a significant time input)"
470  l.45 There is no 3.8.1 in this reference (reference currently points to Chapter 4, if you
target 3.8.1 consider citing Teske et al.)
   o  Corrected as advised
l.60 Please specify what "this" refers to
   o  Now "Silicone"
475  l.64 "suite of tools" are these all python tools?

          ○   Indeed, "Python" added

l.66 Are there more than these two? Else consider to add "and/as well as" before "harmonise..."

    •   The pipeline can be considered to include the OpenSCM part, although this isn't strictly developed by the IAMC, it's therefore a little fuzzy.

          ○   "This pipeline includes tools to manipulate and plot IAM data (pyam, (Gidden & Huppmann, 2019)) and harmonise mismatches in historical emissions (aneris, (Gidden, Fujimori, et al., 2018)). The estimation of climatic impact is performed by OpenSCM, managed by the OpenSCM community (Nicholls, Gieseke, Lewis, & Willner, 2020), which is compatible with the data structure of the pipeline."

l.67 Consider deleting ", managed by the OpenSCM community"

    •   As above, we have revised the sentence slightly according to this comment, while the text still makes clear that this is not part of the IAMC

l.82 several? Three/two: : :

    •   Currently 4 (one due to an update from the original paper draft), more may well emerge. One is arguably an "aggregation tool" rather than a "multiple infiller". We maintain ambiguity for this reason

l.85 Maybe change to "Currently, there are ..."?

          ○   "Currently" added

l.98 Consider rephrasing e.g. "... and where emissions are expected to scale with each other ..."

          ○   Rephrased, this discussion is moved to the next paragraph. "The ratio-based approaches are better for cases where the lead values to be infilled are outside the range in the infiller database and we expect the emissions to scale with each other."

l.98 What do you mean with "regional data" and "aggregate data"? If you refer to regions as subset of global data then this is the first time that a spatial reference is given and I wonder if it would be appropriate to introduce this more formally earlier in the manuscript?

          ○   Reference to regional data has been removed as discussed above. A reference to "splitting up an aggregate basket of emissions" mentioned above, this case now changed to "splitting up aggregated emissions into their components"

l.100 Consider to give an example? CO2 uptake?

          ○   Added ", e.g. $CO_2$ emissions"

l.106 What do you mean with similar – similar magnitudes?

          ○   Indeed, clarified to ", preferably with both larger and smaller lead emissions in the infiller database"

l.111 "estimate the ratios" – if not to be predefined: : :

          ○   Now "determine"

l.112 "follower value in infillee database" -> "in the ..."

          ○   Added as requested

l.112 Please consider to visually separate (and number) the equations.

          ○   Change made for all significant equations

l.114 "each different timeseries" -> different regarding what - do you mean different follower variables?

          ○   Sentence removed.

l.116 mean regarding what - time or different sources (models, scenarios?)

          ○   Sources, clarified to "The 'latest time ratio' method uses the ratio between the mean follower data in the infiller database (we denote this database with lower-case, $e_f$) and the value of the lead variable in the target data ($E_l$)"

l.117 what does "both" refer to?
- o Both means follow and target lead. Now "both values"

l.119 what do you mean with "all estimates"? Different sources (models, scenarios)?
- o Yes, specification of "at that time" added

l.120 why historical? Couldn't this also be different scenarios from different IAMs?
- It can, it's simply that this is the most common use-case (we are yet to see a case where all IAMs model a gas up to a given point then all stop thereafter whilst continuing to model other gases).

l.120 "and the lower case ef (t) represents the follower values in the database at time t." -> "and the lower case $e_f(t_{last})$ represents the follower values in the database."?
- o Rephrased as requested, the distinction being between *E* and *e*.

l.125 the infillee lead is not part of the formula – is this the El(t) from l.113?
- o Yes, it will be used to multiply the ratio.

l. 125-133: Maybe consider to restructure? You could start with the context, i.e. the algorithm name, and then add the explanation, e.g.: "The decompose collection multiple infiller is based on: : : relying on the useful property of : : :"
- o Significantly restructured and added more equations as explanation

l.134-137: Equation for R(t)?
- We explain that this follows the logic below instead, it would be tedious to write out here again with one symbol different. The whole passage has been rewritten.

l.142 I did not understand this sentence
- o Rephrased to "This is the only infiller that is not time-independent, i.e. changing the value of the lead at one time may result in different outputs at other times."

l.143-145: Equation for Ef?
- o We have added it.

l.151 There is no 3.8.1 in that reference (reference currently points to Chapter 4, if you target 3.8.1 consider citing Teske et al.)
- o Well spotted, resolved as above

l.152 There seems to be a lost copy of the figure caption in the text. (Either just delete or maybe rephrase to steps with complete sentences)
- o Indeed, deleted

l.155-165 Equation for Ef?
- This would be illegibly long and complex without a great deal of separately defined objects. The whole section has been rewritten however.

l.163 Maybe Ef not El?
- The weights are associated with e_f, the ordering is associated with e_l. This has been rewritten, as noted above.

l.166 What about the KyotoGHGs as one basket?
- Could be considered (and can be calculated by the code in the examples github repository if you're interested), but interpreting it is complicated by the fact that
- 0) calculating the Kyoto GHGs as a single basket only makes sense if all relevant Kyoto gases are reported, which isn't often the case (if it were Silicone's domain of applicability would be severely reduced)
- 1) there are several possible metrics for this, on top of which the values stored in the IIASA database for a given metric do not entirely equal the values calculated from their components. Sometimes this is due to the incompleteness issue, other times there appear to have been rounding errors or method disputes in the process.
- 2) it naturally correlates very well with its primary components $CO_2$ and methane without really signifying anything, further complicating the averages analysis.

l.169 Maybe give an example for two such variables?

    o   Added "for instance black carbon and carbon monoxide are both produced by incomplete combustion"

l.185 Consider explicitly listing the two constituents

- OK, it is somewhat long-winded to spell out the acronyms here, so we simply use them and hope those unfamiliar with them will look at the table

    o   "(AFOLU and Energy and Industrial processes, a similar concern can be raised about F-gases)"

l.186 Which are "these two" – CO2 and CH4?!

    o   Yes, spelt out now. "$CO_2$ and $CH_4$"

l.188 you write that BC, CO and OC "correlate poorly with others, however, from the table it seems that they do not correlate less well with others than other gasses, the main difference is that they correlate very well with each other.

    o   Clarified: "that correlate well with each other but less well with other emission pathways"

l.190 maybe aggregate F-gas emissions / F-gasses as a basket?

    o   As a basket added to the analysis

l.191 up to here always "CH4"

    o   Changed to always be $CH_4$

l.198 consider deleting "and find similar results"

    o   Ok

l.199 "we choose four" – which are basically all?

    o   Following from the point below, this now reads: "We use the crunchers that are designed for use on complete datasets with only default settings: QRW (default settings mean in absolute mode and for the 0.5 quantile), RMS closest, EQW, time-dependent ratio and linear interpolation. Interpolate selected model behaves identically to linear interpolation with default settings and is not treated separately here."

l.199 But even if there are errors, wouldn't it be interesting to see what happens? I would appreciate if you could also show results for the 'time dependent ratio' algorithm

    o   We now show the results from using the time-dependent ratio too, and slightly change our normalisation to make the comparison with "just use the mean" clearer. In several cases it does indeed give values worse than 1 (i.e. using the mean). A complicating consequence of this new normalisation and higher possible values is a strong skewing of the results and non-normal distribution of the errors. We therefore also substitute the Wilcoxon t-test for the student t-test in all cases. The statistics are all robust to this and it has no impact on our conclusions, although all of the p-values are slightly different now. References to Time-dependent ratio have also been added to the conclusion here.

l.201-202 Equation?

    o   The (new) equation has been added: "i.e. $\langle\sqrt{\langle\left(\frac{E_{f,inf}-E_{f,act}}{\sigma}\right)^2\rangle_i}\rangle_{decade}$, with the subscript text $inf$ indicating that the value is infilled, $act$ indicating actual and $i/decade$ indicating averaging over model/scenario cases or decades."

l.203 "both cases" – CO2 and CH4?

    o   Indeed, clarified

l.203 "non-CO2 pathways" – but CO2 is derived with CH4 -> maybe replace by "emission pathways"

    o   We have replaced this with "follower pathways"

l.204 I would rephrase this, if QRW would be fairly similar all four would be?

    o   Rephrased as "the next smallest"

l.208 What do you conclude from the non-Gaussian distribution test?

610    - o Originally we concluded that we could use the Student's t-test. With the new normalisation and data, we now fail this test so use the Wilcoxon t-test. The qualitative conclusions are as before.
l.212 "either of CO2 or CH4"
    - o Corrected to "either $CO_2$ or $CH_4$"
l.214 capital T for Table 3
615    - o Corrected
l.225 Add Silicon -> "Data in the Silicon package"
    - o Clarified to "in the Silicone examples package", the main Silicone package is not dependent on any fixed data source. (Other data sources are used by the examples package, but are also included in it.)
l.238 treatment of regions has not been introduced, maybe explain better or consider
620 deletion?
    - o Regions are now mentioned above, but not here, as described above.
l.243 "this database" – which? The SR1.5 repository?
    - o This whole section has changed, the data here now comes from a different (named) database, AR5.
l.249-252: Again a lost copy of a figure caption
625    - o Deleted
l.285-287 "free variables" are mentioned twice but are not further explained?
    - o They are not tremendously relevant to the discussion here, but this has been changed to "in this case, based on rates of transition between the RCP pathways and a long-term emissions value"
l.287-288 I do not understand this sentence
630    - o Changed to "Silicone provides an alternative means of complementing such results – instead of specifying the functional forms of all emissions, you can have a few key emissions prescribed and infill the remainder using scenarios with similarities to the desired narrative."
l.300 consider deleting "of which there are many" or maybe replace by "several options"
or the like
635    - o Removed
Table2: Please explain the asterisk again in this figure caption
    - o Added "Names followed by asterisks use a ratio-based approach, i.e. they find a multiplicative factor and then multiply the target lead by this value, if the asterisk is in brackets there are ratio-based options."
Table3: Consider to change the colouring – to me the yellow/orange highlighting gives
640 a 'positive' impression. Maybe you could colour the cells with bold numbers in green
and those which are currently yellow in red?
    - o The table has been recoloured. We have avoided using green and red for colourblindness reasons, but yellow for strong correlation and blue for high time-variability should perform the same roles.

[revised manuscript text omitted]

where subscript < or > signs indicate the model/scenario combination with lead values immediately below or above the target lead value at that time. If multiple points have exactly the same lead value, the average follow value is used. The follower value returned is then the interpolated value for the target lead. The 'Interpolate specified scenarios and models' cruncher filters for scenarios and models that match a given text string before performing the same action of the

875 linear interpolation cruncher.

The 'quantile rolling windows' cruncher, applied with the default option 'use_ratio=False', infills the values based on interpolating between the required quantile of the follower variable. This is calculated at fixed points across the range of

880 lead values in the infiller database for each time. The process is identical to the above discussion where 'use_ratio' is True, except using the actual follower values instead of the ratios between lead and follow. It is inspired by the Generalized Quantile Walk approach in section 3.8.1 of (Meinshausen and Dooley, 2019). An illustration of the idea behind this cruncher

is shown in ~~Figure 2: Schematic of how the quantile rolling window cruncher determines the follow value to use. a) Example relationships between lead (CO₂) and follow (CH₄) variables over time. b) Five windows are drawn and a weighting function constructed for each window. c) A relationship between the sum of the weights and the follow value is established and the follow value at the desired quantile is returned.-~~ $e_p$, including the highest and lowest values. For each window, the weightings of each point are given as

$$w_p(e_l(i)) = 1/(1 + \left(\left(e_p - e_l(i)\right)\big/ d_l\right)^2 ),$$
(8)

 where $d_l$ is the decay length, which defaults to half the separation between $e_p$, and $i$ the label for which model/scenario we are investigating. Increasing the decay length will reduce the weight difference between points, so the rolling window becomes wider and more even, with the limit case of calculating quantile $q$ of all data for large $d_l$. Amongst other things, this is a clear improvement over the Generalized Quantile Walk approach, as the latter uses equal weights within a fixed window of a certain fraction of the infiller database's lead values in a certain year. These values are then normalised so that $\sum w_p = 1$ and sorted into ascending order by $e_f$. The follow value  at quantile $q$, evaluated at lead point $e_{lp}(j)$, is where the quantile equals the sum of weights of all smaller $e_f$ plus half the weight of $e_f(j)$ itself. Note that we sum over smaller *follower* values, but the weighting is determined by the *lead* values:

$$q\left(e_l(j)\right) = \sum_{e \neq f_i(i) < r_p e_f(j)} w_p\left(e \pi_{li}(i)\right) + \frac{w_p\left(\tau e_{lp}(j)\right)}{2}.$$
(9)

[revised manuscript text omitted]
 that with this fairly large infiller database that in both for both $CO_2$ and $CH_4$eases with this fairly large infiller database, the approach that generates non-$CO_2$ follower pathways most similar to those removed from the initial scenarios (i.e. the smallest errors) is the RMS technique, with and that the the QRW technique being is the next smallestfairly similar. Linear interpolation without smoothing is expected to produce a noisy fit when given a large input infiller dataset, so its performance is unsurprisingly worse. The Equal Quantile Walk (EQW) performs similarly poorly, due to effectively
965 ignoring the relationship between the lead and follower data. The time-dependent ratio method is worst of all – its errors are potentially unbounded and for $CO_2$ the average error far exceeds one. To determine the appropriate statistics to apply on the errors, we first perform Aa Shapiro-Wilks test to detect any non-Gaussian aspect for the error distribution (details can be found in the "statistics_for_paper" notebook of the examples github repository). This indicateds that neither $CO_2$ nor the distributionsmethane are statistically significantly non-Gaussian in distribution, for several crunchers either when analysed
970 separately when analysed separately and most clearly for each cruncher or as an aggregate. We will therefore use nonparametric 
[revised manuscript text omitted]

1225

[Figure]

[Figure]

[Figure]

**Start**

Infill many values in similar ways?

no → Emissions are the sum or difference of known values?

yes → Use **infill all required values**

yes (Emissions are the sum...) → Use **infill composite values**

also specify a cruncher

no → Have follower data?

none → Use **constant ratio**

data before some time → Use **last time ratio**

Expect proportionality?

yes → Use **time-dependant ratio**

no → Breaking aggregate data into parts?

yes, one emission different/much larger → Use **split collection with remainder**

yes, all emissions similar → Use **decompose collection**

no → Want a distribution of values?

yes → Use **quantile rolling windows**

no → Small number of similar scenarios?

yes → Use **interpolate specified**

no → Prefer consistency over time or stability?

stability to small changes → Use **quantile rolling windows**

Consistency → Use **RMS closest**

**Figure 2: Schematic of how the quantile rolling windows cruncher determines the follow value to use. a) Example relationships between lead (CO₂) and follow (CH₄) variables over time. b) A number Five of rolling windows centers (here 5, by default 10) are drawn and a weighting function constructed for each window. It has a continuous distribution, rather than a discrete cutoff, hence the name. c) A relationship between the sum of the weights and the follow value is established and the follow value at the desired quantile is returned.**

[Figure]

1235

**Figure 3: Left: The  POEM scenario B projection for CO₂ from Energy and Industrial Applications data. The fine lines represent the different timeseries in the SR1.5 database used to perform the infilling and are not included in the legend for clarity. Right: The results of interpolating this data using  five different crunchers. The interpolate specified model approach used the MESSAGE model and only choses scenarios based on SSP2 pathways.~~the REMIND-MAgPIE 1.5 model and only choses**
1240 **scenarios based on SSP5 pathways.~~**

[Figure]

**Figure 4: The Climate Action Tracker (CAT) Kyoto gas totals (thick lines) compared with the portfolio of values in the SR1.5 database (thin lines).**

[Figure]

**Figure 5: The CAT Kyoto gas baskets decomposed into their components, using the decompose collection multiple infiller.**

[Figure]

**Figure 6: Kyoto gases, decomposed by first infilling the non-negative emissions using the (non-ratio) quantile rolling windows, then infilling the CO₂ using infill composite values.**

[Figure]

**Figure 7: Illustration of using the interpolate specified scenario cruncher to infill a series of stylised trajectories (solid lines), characterised by two different parameters ($\tau$ and $E_\infty$), defined in** (Sanderson et al., 2016). **The first column compares the total CO₂**

1260 calculated for the stylised trajectories to the values of the MESSAGE model for a given group of SSP scenarios (dotted lines). These are our lead values in each case. The second column shows the range of follow values for that SSP. The third column shows the resultant AFOLU (Agriculture, Forestry and Other Land Use) trajectories that emerge from using the Interpolate Specified Scenario infiller.

[Figure]

1265

---

## Author Response (AR2)

**Second response to reviewer 2**

Many thanks for taking the time to review the work again. We are happy you are satisfied with the progress and believe we can solve the remaining problems. We give our reply as itemised comments below.

There has been one suggestion by reviewer 1, where the reviewer recommended to add a short section on limitations, which so far has not been included in the revision and could probably be of further help for those considering to apply the tool (whereby Table 1 does already cover the pitfalls for the single crunchers in detail).
Apart from this aspect I only have a few specific comments left and I recommend to accept the article provided minor revisions are made.

- Great! Comments were previously added on this matter thoughout the paper, however a specific section has now been added.

   - "Note that all of the methods listed above are purely statistical in nature: if the scenarios in the infiller database are fundamentally different from those in the target database, different relationships are likely and the validity of the results is poor. The adequate use of Silicone requires users to select an infiller database most appropriate for each respective application. Using Silicone with an infiller database that has itself been infilled may distort the model democracy of the results. Note also in version 1.0.0 of Silicone, all methods take only a single lead value, although forthcoming work will add the capacity to use multiple lead values to some crunchers. This will improve the ability to resolve more complex relationships, since it is possible for very different worlds to have similar emission trends in one emission without being similar in other emissions."

l. 663 Sentence structure seem to got mixed up

- It has been reworded for clarity:

   - "Infilling Silicone's infilling capability broadens the range of IAMs available for exploring projections of future climate change, hence Silicone forms part of the open-source pipeline for assessments of the climate implications of IAM scenarios"

l. 732 Missing bracket

- Added, sentence rearranged

   - "(called aneris, (Gidden et al., 2018a))"

l. 740 what do you mean with "the rest"?

- Reworded for clarity

   - "Silicone reproduces one entry in the database given the other entries"

l. 780 e.g. -> for example

- Changed

l. 820 Equation not numbered and R.E should probably be R*E?

- Both changed, later numbers incremented

l. 882 Reference not corrected in this place (Teske et al., 2019)

- Indeed, corrected

l. 933 I would nevertheless replace the abbreviation AFOLU here

- OK, spelt out

l. 983 maybe two exceptions?

- It is a single exception for a pairwise comparison
    - Added "pairwise" for clarity

l. 987 e.g. -> for example

- changed

l. 1014 "supported by to the"?

- "to the" removed

l. 1041 grammar? Maybe "... gas total. Therefore, one of the ..."

- Changed

l. 1091-92 I find it a rather strong statement that Silicone allows to make reasonable climate assessments. I recommend to tune this sentence down and rather state that Silicon can help to broaden the range of IAMs to support climate assessments.

- OK
    - "this paper has demonstrated that Silicone can easily be used to allow the involvement of a broader range of IAMs in making climate assessments."

l. 1205 what do you mean with "ratio-a"?

- Typo:
  - "a" -> "based".

[For some reason the section numbering is registered as a change in spite of being the same as in the original]

**Formatted Table**

[revised manuscript text omitted]

Formatted Table